# TORC2 inactivation promotes heterochromatin formation in rDNA and prolongs viability of quiescent fission yeast cells
Hayato Hirai [1,2] ✉ & Kunihiro Ohta [1,3,4] ✉

A large amount of the energy produced by glucose is consumed in the biogenesis of ribosomes, the cellular machinery for protein synthesis. Recent studies suggest that a low-calorie diet and the suppression of ribosome biogenesis can extend lifespan. However, the molecular mechanisms underlying these phenomena remain elusive. Here, we demonstrate that TORC2 (TOR complex 2) promotes ribosomal RNA (rRNA) transcription by facilitating the association of Paf1C (RNA polymerase II-associated factor 1 complex) with the rDNA region. Under glucose starvation, inactivation of the TORC2–Gad8 pathway leads to the dissociation of Paf1C from rDNA, thereby promoting heterochromatin formation and transcriptional repression. This mechanism is distinct from TORC1-mediated gene regulation of rDNA. Additionally, simultaneous inactivation of the redundant TORC1 and TORC2 pathways in nutrient-rich conditions leads to robust rDNA heterochromatin formation and rRNA transcriptional suppression, which is associated with prolonged viability of quiescent cells. This extension of viability is attenuated by the disruption of the H3K9 methyltransferase Clr4. These results suggest that robust heterochromatin formation in the rDNA region may support sustained survival of quiescent cells.

Ribosomes, the cellular machinery responsible for protein synthesis, are the most abundant organelles in eukaryotic cells, requiring a substantial allocation of cellular resources[1–3]. Therefore, the precise regulation of ribosome biogenesis is crucial for maintaining cellular energy homeostasis, which supports cell growth, proliferation, and survival. The dysregulation of ribosome biogenesis can disrupt cellular energy balance, leading to metabolic abnormalities as well as cancer development caused by aberrant cell proliferation[4,5]. Moreover, mutations in genes associated with ribosomopathies, a group of rare diseases, affect components of the ribosome or factors involved in its biogenesis. Such defects impair proper ribosome assembly and maintenance, particularly in hematopoietic cells, resulting in anemia and bone marrow failure[6].

Conversely, there are circumstances in which the flexible regulation of ribosome biogenesis becomes critical, particularly during nutrient starvation. In such states, cells must conserve intracellular resources to maintain viability until nutrient availability is restored. To achieve this adaptation, the energy-intensive process of ribosome biogenesis must be rapidly and efficiently downregulated[7–10]. The Target of Rapamycin (TOR) signaling pathway plays a pivotal role in connecting nutritional status to cellular growth and metabolism. This pathway comprises two distinct complexes, TORC1 and TORC2[11,12]. In response to nutrient starvation, TORC1 inactivation suppresses the transcription of genes encoding ribosomal RNA (rRNA) and ribosomal proteins (RPs), thereby halting ribosome biogenesis[13]. Although this regulatory mechanism has been extensively studied in the budding yeast *Saccharomyces cerevisiae*, its conservation across diverse organisms remains unclear[14].

In the fission yeast *Schizosaccharomyces pombe*– which, unlike *S. cerevisiae*, possesses RNA interference (RNAi) machinery, the histone methyltransferase Clr4, and an HP1-family protein[15]–we recently showed that these factors contribute to heterochromatin formation in the rDNA

[1]Department of Life Sciences, Graduate School of Arts and Sciences, The University of Tokyo, Meguro-ku, Tokyo, Japan. [2]Department of Basic Medical Sciences, Tokyo Metropolitan Institute of Medical Science, Setagaya-ku, Tokyo, Japan. [3]Universal Biology Institute, The University of Tokyo, Bunkyo-ku, Tokyo, Japan. [4]Collaborative Research Institute for Innovative Microbiology, The University of Tokyo, Bunkyo-ku, Tokyo, Japan. ✉e-mail: hirai-hy@igakuken.or.jp; kohta-pub2@bio.c.u-tokyo.ac.jp

region during glucose starvation, thereby repressing rRNA transcription[10]. This starvation-induced heterochromatin formation, triggered by TORC1 inactivation, involves the altered localization of key factors, including the acetyltransferase Gcn5, the stress-responsive transcription factor Atf1, and the histone chaperone FACT[16]. These findings demonstrate the critical role of TORC1 in linking nutrient availability to ribosome biogenesis through the transcriptional control of ribosome-related genes and rRNA.

In contrast, the role of TORC2 in ribosome biogenesis is not well understood. TORC2 is primarily recognized for its roles in regulating the actin cytoskeleton and maintaining genome stability[17–20]. RNA-seq analyses of TORC2-deficient strains have revealed changes in gene expression near sub-telomeric regions but have shown no significant impact on ribosome-related gene expression[21,22]. Nevertheless, because the transcriptional regulation of ribosome-related genes and rRNA is governed by distinct mechanisms[13,14,23], it remains uncertain whether TORC2 directly influences rRNA transcription.

Interestingly, inactivation of the TORC1 pathway in nutrient-rich conditions has been shown to extend lifespan across multiple organisms[24–26]. Studies employing Rapamycin treatment or TORC1-inactivating mutants suggest that reduced TORC1 activity in nutrient-rich conditions promotes longevity through mechanisms including the activation of autophagy, the suppression of protein synthesis, and the regulation of mitochondrial function[27,28]. However, as TORC1 inactivation simultaneously affects numerous cellular processes, the process most directly linked to lifespan extension is unclear. In the case of TORC2, the effects on chronological lifespan in fission yeast appear to be variable and highly context-dependent, with both lifespan extension and shortening reported under different culture conditions[25,29,30]. Importantly, the specific molecular mechanisms by which TORC2 dysfunction affects lifespan remain unclear.

In addition, the age-associated loss of heterochromatin has been proposed to drive aging by impairing gene silencing[31]. For example, flies with impaired heterochromatin formation exhibit approximately twice the pre-rRNA levels compared to those in the wild-type strain, correlating

with a shortened lifespan[32]. Similarly, heterochromatin loss has been observed in Werner syndrome, an adult-onset progeroid disorder, suggesting that heterochromatin maintenance may be a critical determinant of human aging[33]. However, whether TORC1-mediated regulation of heterochromatin formation in the rDNA region directly influences lifespan remains unclear.

This study demonstrates that TORC2, acting independently of TORC1, contributes to the control of ribosome biogenesis by regulating rRNA transcription through heterochromatin formation in the rDNA region. Furthermore, we show that the inactivation of both TORC1 and TORC2 in nutrient-rich conditions enhances rDNA heterochromatinization and suppresses rRNA transcription, which is associated with prolonged maintenance of viability in quiescent cells.

## Results
### TORC2 accumulates in the rDNA regions and promotes rRNA transcription

Between the two types of TOR complexes, we have previously demonstrated that TORC1 localizes to the rDNA region and promotes rRNA transcription[16]. To determine whether TORC2 has a similar function, we hypothesized that TORC2 also accumulates in the rDNA regions. To investigate this, we tagged the C-terminus of the TORC2 component Tor1 with a FLAG epitope and examined the binding of Tor1-FLAG to rDNA using chromatin immunoprecipitation (ChIP). In fission yeast, rDNA is located at both ends of chromosome III, where a unit composed of the 5'ETS, 18S, ITS1, 5.8S, ITS2, 28S, and 3'ETS regions is tandemly repeated approximately 100 to 150 times[34] (Fig. 1a). Tor1-FLAG was broadly enriched across the rDNA unit, spanning the 18S, 5.8S, 28S, and 3'ETS (Fig. 1b). It has been reported that TORC2 activity is suppressed by glucose starvation[20]. When cells were exposed to glucose starvation, which suppresses rRNA transcription, the accumulation of Tor1-FLAG at the 18S, 5.8S, and 28S regions was reduced, despite no changes in the total cellular level of Tor1-FLAG protein (Fig. 1b, c, and Supplementary Fig. 1). These

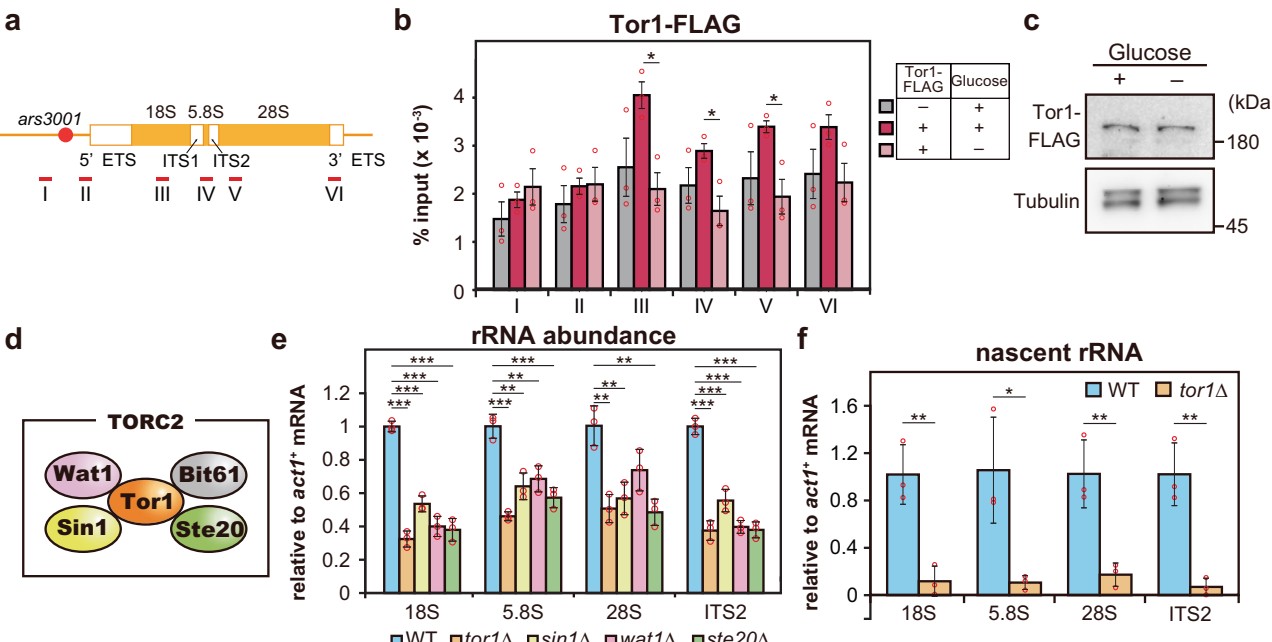

**Fig. 1 | TORC2 accumulates in the rDNA regions and promotes rRNA transcription. a** Schematic diagram of the fission yeast rDNA locus. Red lines with numbers indicate the positions of primers used for ChIP-qPCR. **b** ChIP-qPCR analysis showing the IP/input of Tor1-FLAG in the rDNA region under glucose-rich (dark pink) and glucose-poor (light pink) conditions. Endogenous Tor1 without an epitope tag was used as a negative control (gray). Bars represent means ± SEM, n = 3 experiments. **c** Immunoblot showing the protein levels of Tor1-FLAG and α-tubulin before and after glucose starvation. **d** A schematic diagram of the components constituting TORC2. **e** rRNA levels in WT, *tor1Δ*, *sin1Δ*, *wat1Δ*, and *ste20Δ* strains, normalized to *act1* mRNA levels. Bars represent means ± SD, n = 3 experiments. **f** Newly synthesized rRNA levels in WT and *tor1Δ* strains, normalized to *act1* mRNA levels. Bars represent means ± SD, n = 3 experiments. *p*-values were calculated by Student's *t*-test. ***p < 0.001, **p < 0.01, *p < 0.05.

findings suggest a potential role for TORC2 in contributing to rRNA transcription through its association with the rDNA loci.

If TORC2 dissociation from rDNA contributes to the regulation of rRNA transcription, the loss of TORC2 function would be expected to alter rRNA transcription levels. To test this hypothesis, we performed RT-qPCR to assess rRNA abundance in mutants lacking each TORC2 component, including Tor1, Wat1/Lst8, Sin1, and Ste20/Rictor[29,35,36] (Fig. 1d). Our analysis revealed that 18S, 5.8S, and 28S rRNA levels were reduced in all these mutants. Additionally, the abundance of ITS2-containing precursor rRNA was also decreased, suggesting that rRNA transcription itself was impaired in the absence of TORC2 (Fig. 1e). To further confirm this, we labeled newly synthesized RNA with thiouridine, followed by biotinylation for purification with streptavidin beads. In the *tor1Δ* mutant, the amount of nascent rRNA was substantially lower than that in the wild-type (Fig. 1f). These results support the idea that TORC2 may promote rRNA transcription at least in part by accumulating in the rDNA loci, although we cannot exclude the possibility that cytoplasmic functions of TORC2 also contribute to this regulation.

### Loss of TORC2 function promotes heterochromatin formation in the rDNA region

One possible explanation for the reduced rRNA transcription observed upon TORC2 inactivation is a decrease in rDNA copy number. To investigate this, we quantified the rDNA copy number relative to genomic DNA using quantitative PCR. The *tor1Δ* mutant exhibited an approximately 20% reduction in rDNA copy number compared to the wild-type (Supplementary Fig. 2). However, this modest reduction in rDNA copy number alone is unlikely to fully explain the substantial decrease in nascent rRNA synthesis, which was reduced to around 10% of the wild-type level (Fig. 1f). Therefore, we considered that additional factors may contribute to the downregulation of rRNA expression.

It is well established that chromatin structure influences gene expression levels. For instance, from fission yeast to higher eukaryotes, heterochromatin formation characterized by H3K9 methylation is known to repress gene expression[37,38]. Consistent with this, previous studies have shown that heterochromatin formation in the rDNA region suppresses rRNA transcription under nutrient starvation[10,16,39]. These findings suggest that loss of TORC2 function might promote heterochromatin formation in the rDNA region. To investigate this, we examined H3K9 di- and tri-methylation levels by ChIP-qPCR. Our results showed that H3K9me2 and H3K9me3 levels were elevated across the entire rDNA region in the *tor1Δ* mutant (Fig. 2a, b). Notably, the total histone H3 levels in the rDNA region were comparable between the wild-type and the *tor1Δ* mutant, suggesting that the proportion of H3K9me2/me3 was specifically increased (Fig. 2c).

Chromatin compaction is mediated by Swi6/HP1 binding to H3K9me3[40–42]. To determine whether the amount of Swi6 increases in the rDNA region, we fused GFP to the N-terminus of Swi6 and monitored its localization in cells. Since rDNA is located within the nucleolus, we tagged the nucleolar factor Gar2 with mCherry at its C-terminus to visualize the nucleolar region. GFP-Swi6 signals localized at or within the boundary of the Gar2-mCherry signal were defined as being associated with rDNA, and we quantified the number of such GFP-Swi6 foci (Fig. 2d, e). In the *tor1Δ* mutant, the number of GFP-Swi6 foci observed at the nucleolar periphery and inside the nucleolus was slightly increased compared to the wild-type (Fig. 2f, g). Furthermore, ChIP-qPCR revealed that rDNA fragments were more frequently immunoprecipitated with GFP-Swi6 in the *tor1Δ* mutant than in the wild-type (Fig. 2h). These results suggest that the loss of TORC2 function enhances heterochromatin structure in the rDNA regions.

### Loss of Gad8 function induces heterochromatin formation in the rDNA region and suppresses rRNA transcription

TORC2 phosphorylates the AGC kinase family protein Gad8[43]. The phenotypes observed in the *gad8Δ* mutant closely resemble those in the *tor1Δ* mutant, suggesting that most physiological roles of TORC2 are mediated by its downstream factor Gad8[21,43]. Thus, it is plausible that the TORC2-

mediated regulation of rRNA transcription is also governed by Gad8. To investigate this, we first examined whether Gad8 accumulates in the rDNA region. Like Tor1-FLAG, Gad8-FLAG was enriched across the entire rDNA region, particularly in the 18S, 5.8S, and 28S regions. Notably, this accumulation was abolished under glucose starvation without a reduction of protein levels (Fig. 3a, b and Supplementary Fig. 3a).

Additionally, the enrichment of Gad8-FLAG in the rDNA region was reduced in the *tor1Δ* mutant despite total Gad8-FLAG protein levels remaining unchanged, indicating that TORC2 is required for Gad8 localization in the rDNA region (Fig. 3c, d and Supplementary Fig. 3b). Consistent with the *tor1Δ* mutant, the *gad8Δ* mutant exhibited increased H3K9me2/me3 levels in the rDNA and decreased rRNA transcription (Fig. 3e–g). Taken together, these findings suggest that disruption of the TORC2-Gad8 pathway promotes heterochromatin formation in the rDNA, thereby suppressing rRNA transcription.

### Dissociation of Paf1C from rDNA in coordination with Gad8 triggers heterochromatin formation

Our previous studies have demonstrated that rDNA heterochromatin formation under glucose starvation is initiated by the dissociation of the acetyltransferase Gcn5 and the stress-responsive transcription factor Atf1 from rDNA. This is followed by the accumulation of histone chaperone FACT, both processes being induced by TORC1 inactivation[10,16]. Therefore, it is possible that dysfunction of the TORC2-Gad8 pathway might affect the localization of these factors in the rDNA region. To test this hypothesis, we performed ChIP assays targeting Atf1 and Gcn5, the latter tagged with an HA epitope at its C-terminus. The results revealed that the localization of Atf1 and Gcn5-HA in the rDNA remained largely unchanged among the wild-type, *tor1Δ*, and *gad8Δ* mutant strains (Supplementary Fig. 4). These findings suggest that the TORC2-Gad8 pathway regulates rDNA chromatin structure through mechanisms distinct from those mediated by the TORC1 pathway.

It has been reported that Gad8 associates with Paf1C (RNA polymerase II-associated factor 1 complex), a complex known to promote histone turnover and suppress heterochromatin formation[44,45]. Based on this, we hypothesized that inactivation of TORC2-Gad8 pathway may impair the function of Paf1C, thereby promoting heterochromatin formation in the rDNA region. To investigate this, we examined whether the localization of Leo1, a Paf1 component[46], is altered in the rDNA region under conditions where TORC2-Gad8 signaling is downregulated, such as during glucose starvation. We tagged the C-terminus of Leo1 with FLAG (Leo1-FLAG) and quantified its accumulation across the rDNA. We found that Leo1-FLAG within the 18S, 5.8S, 28S, and 3'ETS regions was significantly reduced under glucose starvation (Fig. 4a). Notably, the pattern of Leo1-FLAG localization in rDNA closely mirrored that of Gad8-FLAG (Fig. 3a), suggesting that the recruitment of Paf1C to rDNA depends on Gad8. Supporting this hypothesis, the accumulation of Leo1-FLAG in rDNA was reduced in the *gad8Δ* mutant (Fig. 4b). Taken together, Paf1C dissociation from rDNA under glucose starvation or in the absence of Gad8 may reduce histone turnover, thereby promoting heterochromatin formation.

To investigate whether the loss of Paf1C function promotes heterochromatin formation in the rDNA region, we analyzed H3K9 di- and tri-methylation levels in the *leo1Δ* mutant. We found a significant increase in H3K9me2 and H3K9me3 levels in the 18S, 5.8S, and 28S rDNA regions in the *leo1Δ* mutant compared to the wild-type (Fig. 4c, d). Furthermore, rRNA levels in the *leo1Δ* mutant were reduced relative to those in the wild-type (Fig. 4e). These findings raise the possibility that the TORC2-Gad8 pathway suppresses heterochromatin formation, potentially through promoting the association of Paf1C with rDNA loci.

### Inactivation of both TOR pathways leads to severe heterochromatin formation in rDNA and suppression of rRNA transcription

Our findings suggest that heterochromatin formation in rDNA is independently regulated by the TORC1 and TORC2-Gad8 pathways. Based on

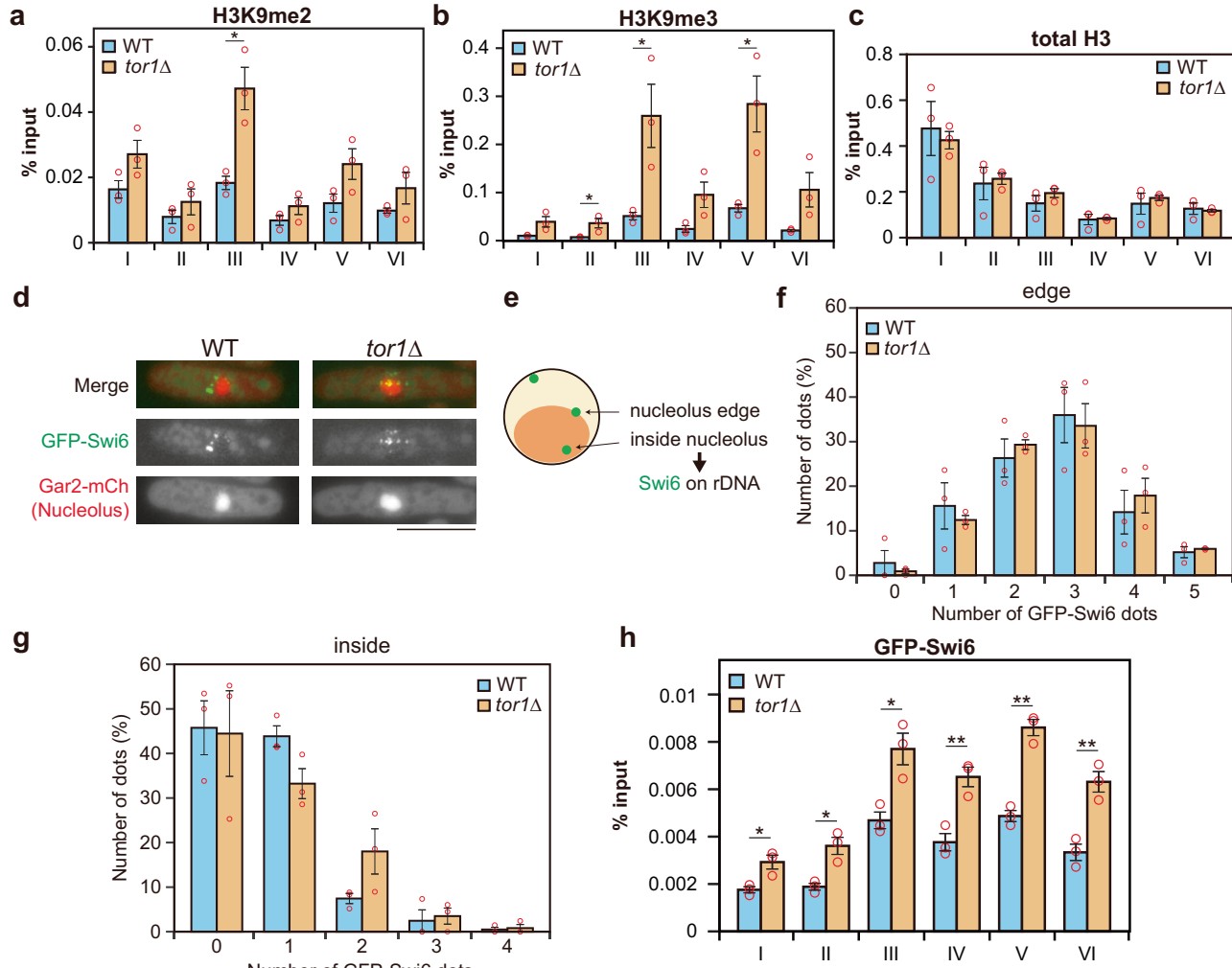

**Fig. 2 | Loss of TORC2 function promotes heterochromatin formation in the rDNA region.** ChIP-qPCR data showing IP/input ratios for H3K9me2 (**a**), H3K9me3 (**b**), and total H3 (**c**) in WT and the *tor1Δ* mutant. Bars represent means ± SEM, n = 3 experiments. **d** Images showing GFP-Swi6 (green) and Gar2-mCherry (red) in WT and the *tor1Δ* mutant. Scale bar = 5 μm. **e** Schematic diagram illustrating the positional relationship between GFP-Swi6 dots and Gar2-mCherry. GFP-Swi6 located at the edge and inside of the nucleolus was defined as localized to rDNA. Number of GFP-Swi6 dots located at the nucleolar edge (**f**) and inside the

nucleolus (**g**) in WT and the *tor1Δ* mutant. Data represent the mean ± SEM of three independent experiments (WT replicate cell counts: 58, 72, and 68; *tor1Δ*: 67, 70, 83). The percentage of cells with ≥ 2 internal GFP-Swi6 foci was significantly higher in *tor1Δ* (23.2%) than in WT (10.6%). Statistical significance was determined by two-sided Fisher's exact test ($p = 0.0007$). **h** ChIP-qPCR showing IP/input ratios for GFP-Swi6 at each point of rDNA in WT and the *tor1Δ* mutant. Bars represent means ± SEM, n = 3 experiments. *p*-values were calculated by Student's *t*-test. ***$p < 0.001$, **$p < 0.01$, *$p < 0.05$.

this, we hypothesized that the simultaneous inactivation of both TOR pathways would result in more severe heterochromatin formation in rDNA and further repression of rRNA transcription. To confirm this, we analyzed the phenotypes of a double mutant harboring the *tor2-287* mutation, which impairs TORC1 activity[47], and the *tor1Δ* mutation. We partially inactivated Tor2 by incubating cells in a glucose-rich condition at a semi-permissive temperature (30 °C) to assess heterochromatin formation in the wild-type, *tor1Δ*, *tor2-287*, or *tor1Δ tor2-287* strains. The partial inactivation of the *tor2-287* single mutant exhibited little or no increase in H3K9 methylation but exhibited reduced rRNA levels in the glucose-rich conditions, suggesting that rRNA transcription can also be regulated by heterochromatin-independent mechanisms. Notably, the *tor1Δ tor2-287* double mutant exhibited the greatest increase in H3K9 methylation and the most pronounced reduction in rRNA levels among all strains tested, surpassing the effects observed in either single mutant (Supplementary Fig. 5a, b).

We also evaluated TORC1 inactivation by rapamycin treatment. In fission yeast, rapamycin can inactivate TORC1 only partially[47], and accordingly, rapamycin treatment alone in the glucose-rich condition influenced neither H3K9 methylation nor rRNA transcription in wild-type

cells. However, rapamycin treatment of *tor1Δ* cells induced much denser heterochromatin in the rDNA region and further suppressed rRNA transcription than in rapamycin-untreated *tor1Δ* cells (Fig. 5a–c). From these results, we conclude that heterochromatin formation in the rDNA is independently regulated by both the TORC1 pathway and the TORC2-Gad8 pathway.

### Inactivation of both TOR pathways is associated with prolonged maintenance of viability in quiescent cells

Previous studies have reported that the inhibition of TORC1 activity in nutrient-rich conditions extends the chronological lifespan of cells[24–26]. However, the precise mechanisms underlying TORC1-mediated effects on long-term survival remain poorly understood. If rDNA heterochromatin formation and suppression of rRNA transcription in nutrient-rich conditions are linked to longer maintenance of cell viability during quiescence, the loss of TORC2 function might also enhance this effect. To investigate this, we measured the viability of cells induced into quiescence, based on the protocol described by the Bähler lab[25] (Fig. 5d). We found that the *tor1Δ* mutant exhibited prolonged maintenance of viability during quiescence

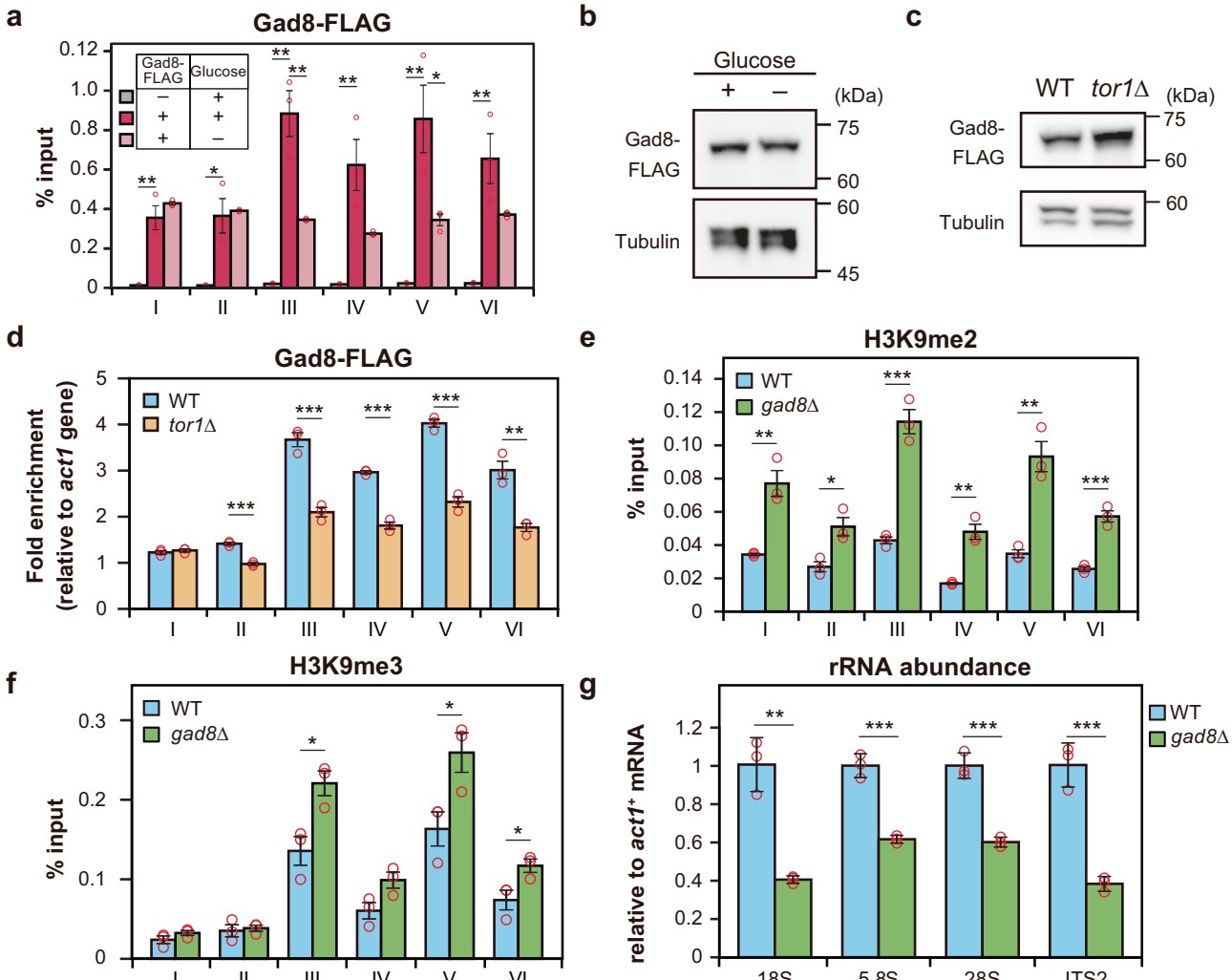

**Fig. 3 | Loss of Gad8 function induces heterochromatin formation in the rDNA region and suppresses rRNA transcription.** **a** ChIP-qPCR data showing the IP/input of Gad8-FLAG in the rDNA region under glucose-rich (dark pink) and glucose-poor (light pink) conditions. Endogenous Gad8 without an epitope tag served as the negative control (gray). Bars represent means ± SEM, n = 3 experiments. Immunoblot analysis of Gad8-FLAG and α-tubulin protein levels under glucose-rich and -poor conditions (**b**) and in WT and *tor1Δ* cells (**c**). **d** ChIP-qPCR analysis showing fold enrichment of Gad8-FLAG in the rDNA region relative to the *act1* gene in WT and the *tor1Δ* mutant. Bars represent means ± SEM, n = 3 experiments. ChIP-qPCR showing IP/input ratios for H3K9me2 (**e**) and H3K9me3 (**f**) in the rDNA region in WT and the *gad8Δ* mutant. Bars represent means ± SEM, n = 3 experiments. **g** Relative amounts of rRNA to *act1* mRNA in WT and the *gad8Δ* mutant. Bars represent means ± SD, n = 3 experiments. *p*-values were calculated by Student's *t*-test. ***$p < 0.001$, **$p < 0.01$, *$p < 0.05$.

compared to the wild-type, although the overall effect was moderate (Fig. 5e). Intriguingly, culturing the *tor1Δ* mutant in the presence of rapamycin led to a further but moderate improvement in the maintenance of viability in quiescence (Fig. 5e). These findings suggest that the simultaneous inactivation of both TOR pathways in nutrient-rich conditions, which is linked to enhanced rDNA heterochromatin formation and reduced rRNA transcription, may be associated with sustained viability in quiescent cells.

To further support this hypothesis, we analyzed the viability of a double mutant lacking the H3K9 methyltransferase Clr4 in combination with the *tor1Δ* mutation in quiescent cells. In the *clr4Δ* single mutant, H3K9 methylation in the rDNA region was completely abolished in nutrient-rich conditions, yet viability over time was comparable to that of the wild-type (Supplementary Fig. 5c, d). This may be because the heterochromatin levels in the wild-type strain under nutrient-rich conditions fall below the threshold required for longer maintenance of viability in quiescence, suggesting that the ability to maintain viability may already be reduced to a minimal level. More importantly, the viability extension observed in the *tor1Δ* mutant treated with rapamycin was largely diminished in the *tor1Δ clr4Δ* double mutant (Fig. 5f). These results indicate that levels of rDNA

heterochromatin formation in nutrient-rich conditions are influenced by the inactivation of TOR pathways, and that these levels may be associated with the capacity to maintain quiescent cell viability over an extended period.

## Discussion
Our findings uncover a pathway by which TORC2 inactivation induces heterochromatin formation at rDNA loci and represses rRNA transcription independently of TORC1 signaling. Mechanistically, the dissociation of Gad8, recruited to rDNA by TORC2, facilitates the release of the Paf1 complex, a key regulator of histone turnover[44,45]. This, in turn, promotes the accumulation of H3K9 methylation mediated by the RNAi-dependent pathway, thereby driving heterochromatin assembly. Notably, dual inactivation of TORC2 and TORC1 synergistically enhances rDNA heterochromatin formation in nutrient-rich conditions and further suppresses rRNA transcription, which is associated with prolonged maintenance of viability in quiescent cells.

Our findings that TORC2-Gad8 binds to rDNA (Figs. 1b, 3a) are consistent with earlier studies reporting the chromatin-binding functions of

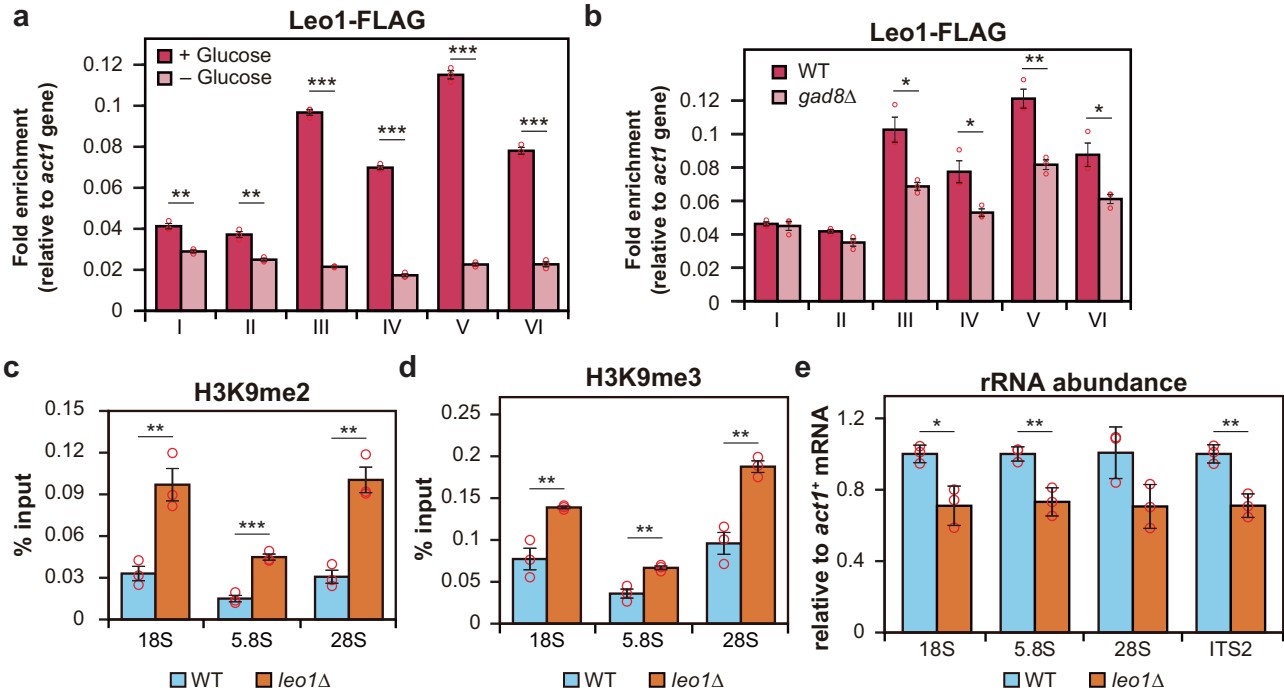

**Fig. 4 | Dissociation of Paf1C from rDNA in coordination with Gad8 triggers heterochromatin formation. a** Comparison of Leo1-FLAG enrichment in the rDNA region relative to *act1* gene under glucose-rich (dark pink) and glucose-poor (light pink) conditions. Bars represent means ± SEM, n = 3 experiments. **b** ChIP-qPCR data showing Leo1-FLAG enrichment in the rDNA region relative to the *act1* gene in WT and the *gad8Δ* mutant. Bars represent means ± SEM, n = 3 experiments. Enrichment levels of H3K9me2 (**c**) and H3K9me3 (**d**) shown as IP/input ratios at the 18S, 5.8S, and 28S regions in WT and the *leo1Δ* mutant. Bars represent means ± SEM, n = 3 experiments. **e** Quantification of total rRNA levels normalized to *act1* mRNA in WT and the *leo1Δ* mutant. Bars represent means ± SD, n = 3 experiments. *p*-values were calculated by Student's *t*-test. ***$p < 0.001$, **$p < 0.01$, *$p < 0.05$.

Tor1 and Gad8[48]. In budding yeast, the helix-turn-helix (HTH) domain of Tor1 (TORC1) facilitates rDNA binding, and this structural motif is conserved in Tor2 (TORC2)[49]. Similarly, in fission yeast, Tor2 localizes to rDNA through its HTH domain[16], suggesting that *S. pombe* Tor1 may accumulate in the rDNA through a comparable HTH domain. On the other hand, although TORC2-Gad8 dissociates from rDNA under starvation conditions (Figs. 1b, 3a), Cohen et al. reported that the chromatin-binding abilities of Tor1 and Gad8 do not change substantially under glucose starvation[48]. This discrepancy implies that there may be an rDNA-specific mechanism whereby TORC2-Gad8 dissociates from rDNA in a starvation-dependent manner. For example, small RNAs produced from the rDNA region are known to increase under starvation[50], and this transcriptional process might destabilize TORC2-Gad8 binding at rDNA under starvation conditions.

Paf1C facilitates histone turnover in constitutive heterochromatin regions and heterochromatin islands, thereby preventing excessive heterochromatin formation[44]. In the rDNA region, Paf1C, recruited by Gad8 (Fig. 4b), likely facilitates histone turnover to suppress heterochromatin formation under nutrient-rich conditions. Conversely, under starvation conditions or in the absence of Gad8, Paf1C dissociates from rDNA (Fig. 4a, b), which may enhance the maintenance of methylated histone H3 and promote heterochromatin formation. In contrast, at subtelomeric regions and the mating-type locus, heterochromatin formation is reduced, and gene expression is upregulated in TORC2-Gad8 mutants[22,51], suggesting a distinct regulatory mechanism from that observed in rDNA. It is possible that Gad8 localizes to transcriptionally active chromatin and interacts with Paf1C. In such regions, the absence of Gad8 may allow heterochromatin formation due to loss of this interaction. Alternatively, TORC2-Gad8, which is primarily localized in the cytoplasm, may phosphorylate Paf1C and thereby prevent its localization to subtelomeric or mating-type regions. Upon TORC2-Gad8 inactivation, Paf1C could remain unphosphorylated and thus associate with these regions, where it interferes with heterochromatin formation.

Intriguingly, the *leo1Δ* mutant exhibits increased small RNA transcription exclusively in the rDNA region[44]. Paf1C has been proposed to inhibit the generation of new siRNAs by ensuring proper transcription termination[52]. Accordingly, the dissociation of Paf1C from the rDNA region may strengthen RNAi-dependent H3K9 methylation, thereby accelerating heterochromatin formation.

Notably, the reduction in rRNA levels in the *leo1Δ* mutant is relatively modest compared to that observed in the *tor1Δ* or *gad8Δ* mutants (Figs. 1e, 3g, 4e). This suggests that factors other than Paf1C might be involved in the regulation of rDNA heterochromatin downstream of the TORC2-Gad8 pathway. Indeed, it has been reported that Gad8 interacts with Bdf2, a factor responsible for histone H4 acetylation[45,53]. Thus, the dissociation of Gad8 from rDNA could lead to reduced histone acetylation, which might contribute to the acceleration of heterochromatin formation.

It should be noted that heterochromatin formation in the rDNA region and suppression of rRNA transcription under nutrient starvation are regulated by distinct mechanisms involving TORC1 and TORC2[16] (Fig. 5a–c, and Supplementary Fig. 5). These two pathways appear to play differential roles depending on the type of nutrient depletion. Under glucose starvation, both TORC1 and TORC2 are inactivated[20,54]. Glucose, being the primary energy source for cells, significantly limits ATP production when depleted. Therefore, rapid and robust heterochromatin formation in the rDNA region and suppression of rRNA transcription lead to the cessation of ribosome biogenesis, minimizing cellular resource consumption.

In contrast, nitrogen starvation leads to TORC1 inactivation while maintaining TORC2 activity[54,55]. Notably, although cells are under nitrogen depletion, heterochromatin formation in the rDNA region has been observed[38], suggesting that TORC1 inactivation serves as a primary driver for promoting heterochromatin formation in this context. Nitrogen starvation is a crucial condition for inducing G1 arrest, a prerequisite for initiating meiosis. However, cells lacking TORC2 function fail to arrest in G1 and cannot proceed to meiosis[36,43]. Thus, under nitrogen starvation, the inactivation of TORC1 likely suppresses ribosome biogenesis to conserve

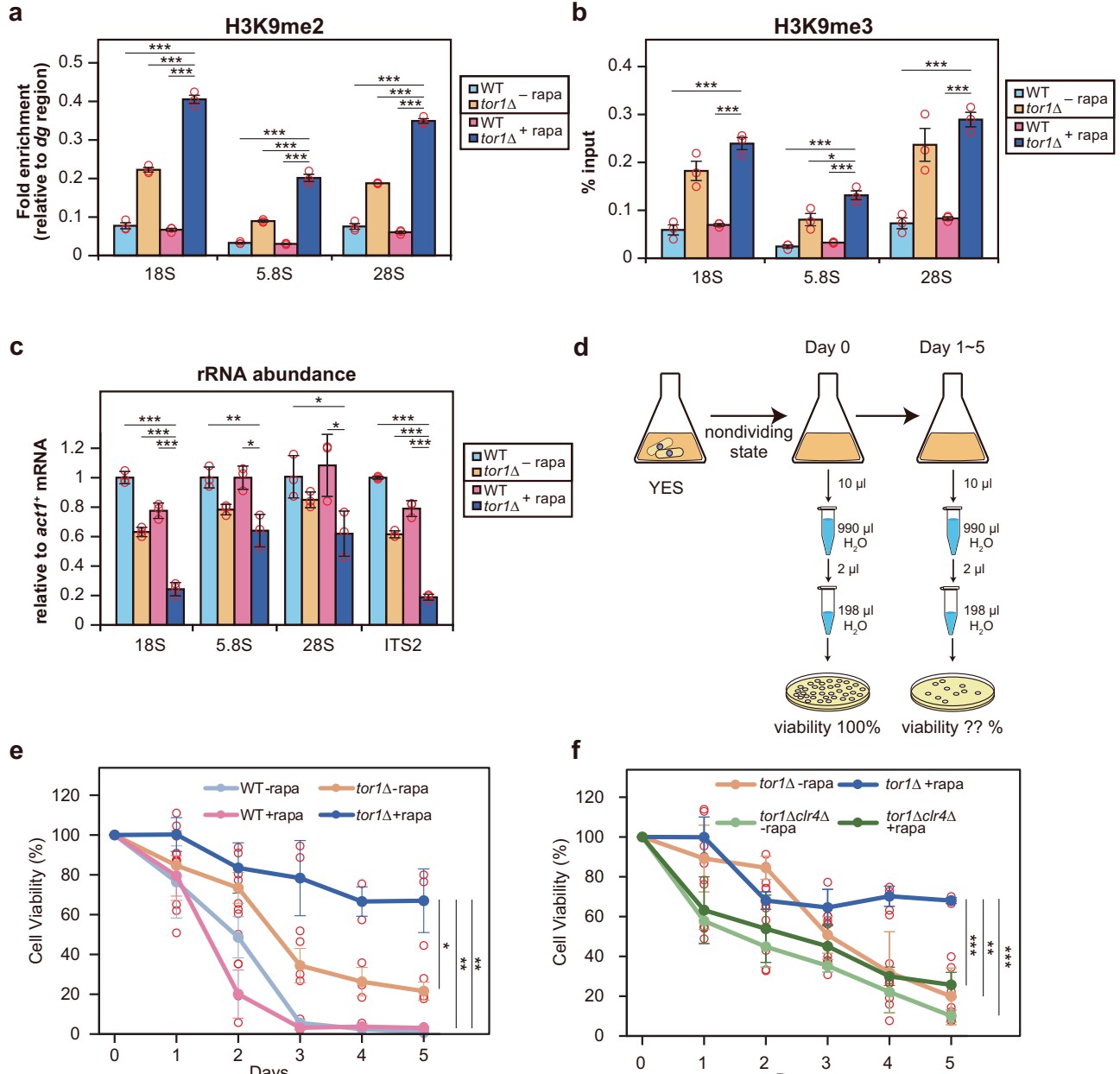

**Fig. 5 | Inactivation of both TOR pathways is associated with prolonged viability of quiescent cells.** Relative levels of H3K9me2 (**a**) and H3K9me3 (**b**) compared to the pericentromeric region (dg) in WT and the *tor1Δ* mutant at 18S, 5.8S, and 28S rDNA loci, with ( + rapa) or without (-rapa) rapamycin treatment. Bars represent means ± SEM, n = 3 experiments. **c** rRNA abundance normalized to *act1* mRNA in WT and the *tor1Δ* mutant with ( + rapa) or without (-rapa) rapamycin treatment. Bars represent means ± SD, n = 3 experiments. **d** Schematic diagram illustrating the method for measuring the viability of quiescent cells. Cells were cultured for 2 days to induce a nondividing state, followed by continuous culturing for an additional 5 days. Equal volumes of cell culture were spread onto agar plates, and the number of viable cells was counted. Viability on Day 0 was used as the baseline (100%). **e** Graph showing the cell viability of WT and the *tor1Δ* cells with ( + rapa) or without (-rapa) rapamycin treatment. Data are presented as mean ± SD, n = 3 biological replicates. **f** Graph showing the cell viability of *tor1Δ* and *tor1Δ clr4Δ* cells with ( + rapa) or without (-rapa) rapamycin treatment. Data are presented as mean ± SD, n = 3 biological replicates. *p*-values were calculated by Student's *t*-test. ***$p < 0.001$, **$p < 0.01$, *$p < 0.05$.

cellular resources, while the sustained activity of TORC2 ensures the progression of the sexual life cycle, including meiosis.

The downregulation of rRNA biosynthesis is associated with lifespan extension in multiple species[56,57]. In this study, we demonstrate that the suppression of rRNA transcription in nutrient-rich conditions, caused by the disruption of the TORC2-Gad8 pathway, is associated with an improvement in survival of fission yeast quiescent cells. Our study further revealed that the simultaneous inactivation of TORC1 and TORC2 in nutrient-rich conditions significantly enhances Clr4-dependent rDNA

heterochromatinization and further represses rRNA transcription, which may contribute to increased survival of quiescent cells (Fig. 5e, f).

However, some studies reported that the disruption of Tor1, a component of TORC2, shortened the lifespan of fission yeast cells[25,29]. This seemingly contradictory result could be attributed to differences in the nutritional composition of the growth medium. In *C. elegans*, it has been shown that the impact of TORC2 signaling disruption on lifespan can vary depending on the feed provided[58]. For instance, while the lifespan of *rict-1* mutants is shortened when fed OP50, a nutrient-rich feed such as

HB101 significantly extends their lifespan[59]. Indeed, in fission yeast, environmental and nutritional differences associated with the type of culture medium have been discussed as potential factors that can lead to opposite effects of *tor1Δ* on chronological lifespan (CLS)[60].

Another possible explanation involves differences in genetic background. While we used strains with a 972 h⁻ background, a previous study employed strains with *leu1-32* background[29]. In our preliminary observations, for unknown reasons, strains with the *leu1-32* background were unable to form heterochromatin at the rDNA region, both under nutrient-rich conditions and under glucose starvation. Alternatively, variations in leucine or other amino acid concentrations in the culture medium may influence rDNA heterochromatin formation in nutrient-rich conditions. Further work is needed to clarify how leucine availability affects rDNA heterochromatin formation and its potential impact on CLS.

Why does the suppression of rRNA transcription lead to prolonged maintenance of viability in quiescence? Recent study has suggested that the suppression of rRNA biosynthesis reduces cellular energy consumption (e.g., ATP), which supports lipidome homeostasis and mitigates mitochondrial stress[61]. Given the decline in intracellular ATP pools with aging[62], preserving ATP levels via rRNA biosynthesis suppression is likely a pivotal mechanism in anti-aging. At the organismal level, on the other hand, the inhibition of rRNA biosynthesis during early developmental stages has been associated with growth delays and an elevated risk of developmental disorders, including ribosomopathies, potentially caused by defective ribosome function[61,63]. Thus, we propose that suppressing rRNA biosynthesis during mid to late developmental stages may represent a promising anti-aging therapeutic strategy, balancing lifespan extension with minimal health risks.

Recent studies have demonstrated that metformin, a drug used to treat type 2 diabetes, possesses anti-aging effects, raising expectations for its potential as an anti-aging drug[64,65]. Metformin is known to activate AMPK, thereby inhibiting the mTOR pathway, which is expected to lead to a reduction in the expression of ribosome-related genes. Indeed, it has been reported that the addition of metformin to the breast cancer cell line MCF-7 reduces rRNA transcription levels[66].

In fission yeast, metformin treatment in nutrient-rich conditions has been reported to extend CLS. Metformin exposure under glucose-starved conditions also markedly prolongs lifespan. These results suggest that inhibition of mTOR by starvation or metformin may underlie this effect[67]. However, other work indicates that the impact on lifespan is not uniform and depends critically on the timing and dosage of treatment[68]. It remains to be explored whether the addition of metformin to primary cultured cells induces heterochromatin formation in the rDNA region and suppresses transcription. This could help elucidate the universality of aging mechanisms and contribute to the advancement of anti-aging research.

Furthermore, studies across various organisms have demonstrated that caloric restriction or short-term intermittent fasting extends lifespan[69]. Recent studies suggest that caloric restriction activates AMPK and that fasting induces nucleolar and chromatin reorganization via the mTOR pathway, leading to the suppression of pre-rRNA transcription[70,71]. However, since the effects of fasting appear to be limited to intestinal cells, it is of interest to determine whether long-term, mild caloric restriction induces chromatin remodeling, particularly rDNA heterochromatinization in cells throughout the body.

## Methods

### *S.pombe* strains and cell culture

The *S. pombe* strains used in this study are listed in Supplementary Table 1. Except for the CLS assay, cells were cultured in YER medium (5 g l⁻¹ yeast extract, 60 g l⁻¹ D-glucose, 100 mg l⁻¹ adenine) at 30 °C for approximately 19 hours. Logarithmic-phase cells were then harvested for subsequent experiments. To induce glucose starvation, the harvested cells were washed twice with YED medium (5 g l⁻¹ yeast extract, 1 g l⁻¹ D-glucose, 3% glycerol, 100 mg l⁻¹ adenine) and cultured for an additional 30 minutes before being harvested. For the experiments shown in Fig. 5a–c, rapamycin was added to the YER medium at a final concentration of 200 ng ml⁻¹ (AG-CN2-0025,

AdipoGen Life Sciences; prepared from a 1 mg ml⁻¹ rapamycin stock solution in DMSO) and cells were cultured for 3 hours before being harvested.

### Yeast genetics

Epitope tagging and fluorescent protein tagging, as well as gene disruption, were performed according to standard protocols[72,73]. Since defects in the TORC2-Gad8 pathway cause G2 arrest, thereby preventing mating[43], gene disruption mutants related to the TORC2-Gad8 pathway were constructed via transformation rather than mating. These strains were selected not only by colony PCR-based genotyping but also by confirming the phenotype of impaired growth at high temperature[74]. Strains with a disrupted *clr4* gene were selected by verifying the loss of H3K9 methylation using ChIP-qPCR, in addition to genotyping.

### Chromatin immunoprecipitation (ChIP)

The ChIP assay was performed according to our previous study[75]. Cells were cross-linked by adding formaldehyde to a final concentration of 1% and incubating for 15 min at 30 °C. The reaction was quenched by adding glycine to 125 mM and incubating a further 5 min at room temperature. Cells were washed twice with TBS, pelleted, and snap-frozen in liquid nitrogen. Pellets were resuspended in 200 μL of buffer I (50 mM HEPES–KOH, pH 7.5; 150 mM NaCl; 1 mM EDTA, pH 8.0; 1% Triton X-100; 0.1% sodium deoxycholate) supplemented with a protease inhibitor cocktail (Roche) and 1 mM PMSF. Cells were disrupted with 0.5 mm zirconia beads using a Multibead Shocker (Yasui Kikai, Japan). Chromatin was sheared on ice with a Handy Sonic sonicator (UR-20P, Tomy Seiko, Japan; six 30-s pulses at power 10 with 1-min intervals) to ~500 bp fragments, followed by centrifugation and collection of the clarified supernatants. For immunoprecipitation, Dynabeads Protein G (10004D, Thermo Fisher) pre-coupled with α-FLAG antibody (1:100, F1804, Sigma-Aldrich) and Dynabeads Protein A (10002D, Thermo Fisher) pre-coupled with α-H3K9me2 (1:1000, ab1220, abcam), α-H3K9me3 (1:1000, 39161, Active motif), α-H3 (1:1000, ab1791, abcam), α-GFP (1:200, 632592, Clontech), α-Atf1 (1:200, ab18123, abcam), or α-HA antibody (1:200, 12CA5, Sigma-Aldrich) were incubated with the chromatin fraction for 3 h at 4 °C. The beads–antibody complexes were sequentially washed on ice twice each with chilled buffer I, lysis 500 buffer (50 mM HEPES–KOH, pH 7.5; 500 mM NaCl; 1 mM EDTA, pH 8.0; 1% Triton X-100; 0.1% sodium deoxycholate), wash buffer (10 mM Tris-HCl, pH 8.0; 1 mM EDTA, pH 8.0; 0.25 M LiCl; 0.5% NP-40; 0.5% sodium deoxycholate), and finally TE. To elute, 50 μL of elution buffer (20 mM Tris-HCl, pH 8.0; 100 mM NaCl; 20 mM EDTA, pH 8.0; 0.1% SDS) was added, and the bead suspension was heated at 65 °C for 15 min; the released protein–DNA complexes were recovered. This elution step was performed twice, and the combined 100 μL eluate was incubated at 65 °C overnight. Protein was then degraded by adding 1 μL proteinase K (20 mg ml⁻¹; Invitrogen) and incubating at 55 °C for 3 h. DNA was purified using the FastGene Gel/PCR Extraction Kit (NIPPON Genetics). Quantitative PCR (qPCR) of the retrieved DNA was performed using THUNDERBIRD® SYBR™ qPCR Mix (QPS-201, TOYOBO) and a StepOnePlus Real-Time PCR System (Applied Biosystems). The primer sets used for qPCR are listed in Supplementary Table 2.

### Protein study

Protein extraction for Western blotting was performed according to our previous study[75]. Cells grown as described above were harvested by centrifugation, washed twice with ice-cold PBS, and snap-frozen in liquid nitrogen. Frozen pellets were resuspended in lysis buffer (50 mM Tris-HCl, pH 7.5; 1 mM EDTA, pH 8.0; 10% glycerol; 150 mM NaCl; 0.05% NP-40; 1 mM phenylmethylsulfonyl fluoride; 1 mM dithiothreitol; protease inhibitor cocktail, Roche). Cells were lysed with 0.5 mm zirconia beads using a Multibead Shocker (Yasui Kikai, Japan). Lysates were clarified by centrifugation and protein concentrations were measured. An equal amount of SDS sample buffer was added to each supernatant, and samples were heated at 100 °C for 3 min. Samples were loaded onto Mini-PROTEAN TGX

precast gels (10%, #4561036, Bio-Rad) and subjected to SDS-PAGE for protein separation. Proteins were transferred from the gel to a PVDF membrane (IPVH304F0, Merck Millipore) using the wet transfer method. To detect the target proteins on the membrane, α-FLAG antibody (1:1000, F1804, Sigma-Aldrich) or α-alpha-Tubulin antibody (1:1000, ab11304, abcam) was used as the primary antibody, and α-mouse IgG-HRP antibody (1:20,000, NA931, Cytiva) was used as the secondary antibody. Chemiluminescent signals were detected using ECL Western Blotting Detection Reagents (RPN2109, Cytiva) and visualized with a LAS4000 mini imaging system (Cytiva). The captured images were processed using Adobe Photoshop (version 2023) to appropriately enhance signal intensity.

## Total RNA extraction

Total RNA was extracted according to our previous study[76]. Frozen cell pellets were resuspended in 250 μL of preheated beads buffer (75 mM $NH_4OAc$, 10 mM EDTA, pH 8.0) and transferred to 1.5 mL tubes containing 200 μL of acid-washed glass beads (G8772, Sigma-Aldrich), 25 μL of 10% SDS, and 300 μL of acid phenol:chloroform (pH 4.5, AM9720, Thermo Fisher Scientific) at 65 °C. The samples were vortexed for 1 min, incubated at 65 °C for 1 min, and this process was repeated three times. After a final 10-minute incubation at 65 °C, the samples were centrifuged at 16,000 g for 15 min at 25 °C. The aqueous phase was transferred to a new tube, mixed with 400 μL of chloroform:isoamyl alcohol (25666, Sigma-Aldrich) and 200 μL of beads buffer, and centrifuged at 16,000 g for 30 min at 4 °C. The aqueous phase was then mixed with 600 μL of isopropanol and 20.4 μL of 7.5 M $NH_4OAc$. Following centrifugation at 20,400 g for 30 min at 4 °C, the RNA pellet was washed with 500 μL of 70% ethanol and air-dried. RNA concentration was measured using a NanoDrop spectrophotometer (Thermo Fisher Scientific), and 1 μg of RNA was used for reverse transcription.

## Isolation of nascent RNA

Nascent RNA was extracted according to our previous study [10,77], as outlined below. To incorporate 4-thiouridine into nascent RNA, cell cultures were supplemented with 4-thiouridine (in DMSO, 16373, Cayman Chemical Company) at a final concentration of 75 μg ml$^{-1}$ and incubated with shaking for 10 minutes. After RNA extraction from the harvested cells, the RNA was incubated at room temperature for 90 minutes in biotin buffer (10 mM Tris–HCl, pH 7.4, 1 mM EDTA, 0.4 mg ml$^{-1}$ EZ-link HPDP-Biotin (21341, Thermo Fisher Scientific)). The biotin-labeled RNA solution was mixed with chloroform:isoamyl alcohol (25666, Sigma-Aldrich) and centrifuged at 20,000 g for 15 min. The aqueous phase was collected, mixed with an equal volume of isopropanol and 1/10 volume of 5 M NaCl, and incubated overnight at −80 °C. Following centrifugation at 20,400 g for 15 min, the RNA pellet was washed with 70% ethanol and air-dried. The biotinylated RNA was resuspended in RNase-free water and conjugated to Dynabeads M-280 streptavidin beads (DB11205, Thermo Fisher Scientific), which were pre-washed three times with MPG buffer (100 mM Tris-HCl, 1 M NaCl, 10 mM EDTA, pH 7.4), and blocked with yeast tRNA (AM7119, Invitrogen) for 30 min. After incubation for 40 min at room temperature, the beads were washed with preheated MPG buffer, followed by incubation at 65 °C for 5 min. This step was repeated three times, and the beads were suspended in 5% β-mercaptoethanol solution and incubated for 5 min. This process was repeated once more to recover RNA separated from the beads. This RNA was further purified using the RNAqueous-Micro Total RNA Isolation Kit (AM1931, Invitrogen) as follows. The RNA solution was mixed with lysis buffer and 100% ethanol, applied to a column, and centrifuged at 20,000 g for 1 min. The column was washed three times with wash solution I and twice with wash solution 2/3. After drying the column, RNase-free water pre-heated to 85 °C was applied to the column, incubated at 85 °C for 1 min, and centrifuged to collect the eluted RNA. The purified RNA was reverse-transcribed as described below.

## Reverse transcription and quantitative PCR

Extracted RNA was reverse-transcribed using the PrimeScript™ RT Reagent Kit with gDNA Eraser (RR047A, TaKaRa) according to the manufacturer's protocol. Quantitative PCR of cDNA was performed using THUNDERBIRD® SYBR™ qPCR Mix (QPS-201, TOYOBO) and a StepOnePlus Real-Time PCR System (Applied Biosystems). Data were analyzed using the ΔΔCt method. The primer sets used in this study are listed in Supplementary Table 2. The $act1^+$ gene was used as a normalization control in RT-qPCR. Previous transcriptome studies have shown that $act1^+$ mRNA levels remain stable in the TORC2 mutant [21,22].

## CLS assay

The chronological lifespan (CLS) of S. pombe cells was measured as described by Rallis et al.[25], with the following modifications. Cells were cultured in 20 mL of YES medium (5 g l$^{-1}$ yeast extract, 30 g l$^{-1}$ D-glucose, 100 mg l$^{-1}$ adenine, 50 mg l$^{-1}$ uracil, 50 mg l$^{-1}$ leucine, 50 mg l$^{-1}$ lysine, and 100 mg l$^{-1}$ histidine) in 100 mL Erlenmeyer flasks, loosely covered with aluminum foil to allow gas exchange. Cultures were incubated at 30 °C with orbital shaking using a bio shaker for 2 days to reach the stationary phase. A 10,000-fold diluted cell suspension was spread onto YES agar plates, and the number of colonies formed was measured to determine the survival rate, with Day 0 set at 100% viability. Subsequently, the number of colonies was measured daily, and graphs visualizing the survival rate were generated. For rapamycin treatment, a rapamycin solution (AG-CN2-0025, AdipoGen Life Sciences) was added to the culture on Day 0 at a final concentration of 200 ng ml$^{-1}$.

## Genomic DNA quantification

Genomic DNA was extracted from approximately $1.5 \times 10^7$ cells using the Dr. GenTLE™ High Recovery for Yeast Kit (9082, TaKaRa) according to the manufacturer's protocol. The extracted genomic DNA was diluted to an appropriate concentration and quantified by qPCR using THUNDERBIRD® SYBR™ qPCR Mix (QPS-201, TOYOBO) and a StepOnePlus Real-Time PCR System (Applied Biosystems). Based on the obtained Ct values, the rDNA copy number of the tor1Δ strain relative to the wild-type was calculated using the ΔΔCt method. The primer sets used for qPCR are listed in Supplementary Table 2.

## Statistics and reproducibility

Statistical analyses were performed using a two-tailed Student's t-test to calculate p-values. p-values are indicated as * (p < 0.05), ** (p < 0.01), and *** (p < 0.001). Sample sizes for each experiment are specified in the figure legends. Reproducibility was confirmed by at least three independent biological replicates.

## Reporting summary

Further information on research design is available in the Nature Portfolio Reporting Summary linked to this article.

## Data availability

The data that support the findings of this study are available from the corresponding author upon reasonable request. The source data are publicly available and can be accessed by anyone from the uploaded Supplementary Data file.

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

## Acknowledgements

We thank the National Bio-Resource Project (NBRP) of MEXT, Japan, for providing the *tor2-287* mutant. This study was supported by Japan Science and Technology Agency (JST) CREST Grant [JPMJCR18S3 to K.O.], Japan Agency for Medical Research and Development (AMED) Grant [JP20wm0325003 and JP22gm1610007 to K.O.], Japan Society for the Promotion of Science (JSPS) KAKENHI [JP23K14175 to H.H.], Institute for Fermentation, Osaka [Y-2024-2-019 to H.H.], Takeda Science Foundation to H.H., and the Dr. Yoshifumi Jigami Memorial Fund, The Society of Yeast Scientists to H.H.

## Author contributions

H.H. conceived and designed this study; H.H. performed the experiments and analyzed data; H.H. drafted the manuscript and prepared the figures; K.O. supervised the project and finalized the manuscript through discussion with H.H. All authors approved the final version of the manuscript.

## Competing interests

The authors declare no competing interests.
