## [Transparent Peer Review file · Communications Biology]

TORC2 inactivation promotes heterochromatin formation in rDNA and prolongs viability of quiescent fission yeast cells

Corresponding Author: Dr Hayato Hirai

Version 0:

Reviewer comments:

Reviewer #1

(Remarks to the Author)

Review of Hirai et al. TORC2 inactivation promotes heterochromatin formation in rDNA to extend the chronological lifespan of fission yeast.

Summary

This very interesting article describes the effect of TORC2 inactivation on rDNA heterochromatin formation in the fission yeast *Schizosaccharomyces pombe*. The authors show that TORC2 inactivation leading to rDNA heterochromatin formation also leads to reduced rRNA transcription and increased chronological lifespan. As multiple studies have suggested in several contexts that reducing ribosomal production is linked to increased lifespan in multiple extremely divergent species, this finding could have a broad impact on our understanding of the basic biology of aging.

The authors first demonstrated that a FLAG tagged version of Tor1, a component of the TORC2 complex, binds rDNA, and that this binding is reduced during glucose starvation, a condition that shows reduced rDNA transcription.

Next, the authors asked whether lack of TORC2 activity altered rDNA transcription by considering mutants lacking the TORC2 components Tor1, Pop3/Lst8, Sin1, and Ste20/Rictor. In each case, these showed reduced levels of rRNA. This was further confirmed by thiouridine-biotin labeling of newly synthesized rRNA in a *tor1Δ* mutant.

Next, the authors considered changes at the rDNA locus. By quantitative PCR they showed a decreased rDNA copy number in the *tor1Δ* mutant, but not one deemed sufficient to fully explain the decreased rRNA. This led the authors to consider heterochromatin formation. They measured H3K9me2 and H3K9me3 levels by ChIP-Seq and found them both increased across the rDNA region in the *tor1Δ* mutant, while the total histone levels appeared unchanged.

To ask whether Swi6/HP1 bound the increased H3K9me3 seen at the rDNA, the authors used a GFP-Swi6 along with an mCherry labeled Gar2, which should localize to the nucleolus, the site of the rDNA. They found a slightly increased GFP/RFP colocalization in a *tor1Δ* mutant relative to wild type.

The authors next focused on Gad8, a downstream target phosphorylated by TORC2, and showed that GAD8 deletion led to increased rDNA H3K9me2/me3 and decreased rRNA similar to *tor1Δ*. They also showed that Gad8-FLAG accumulated at the rDNA and this was reduced by glucose starvation or *tor1Δ*.

While previous work has suggested that Atf1 and Gcn5 dissociation can precede heterochromatin formation at the rDNA, the authors showed that *tor1Δ* and *gad8Δ* do not alter Atf1 and Gcn5 localization, suggesting an independent role for TORC2's effects on rDNA heterochromatin.

As Gad8 has been shown to interact with the Paf1 complex (Paf1C) in telomeric heterochromatin formation, the authors looked at the Paf1C component Leo1 in their context. They showed that Leo1 localizes to the rDNA and this is reduced in *gad8Δ*, and also that *leo1Δ* leads to increased H3K9 di- and trimethylation and to reduced rRNA.

To consider the relationship between TORC1 and TORC2 the authors combined a TORC1 and TORC2 mutant and showed that the increased H3K9 methylation and reduced rRNA effects were both stronger than in *tor1Δ* alone, and that rapamycin treatment in *tor1Δ* also increased these effects relative to *tor1Δ* alone.

Finally, the authors measured the chronological lifespan of *tor1Δ* mutants and found them greatly longlived relative to wild type, and also found that this effect was further increased by treatment of *tor1Δ* with rapamycin. The effect of *tor1Δ* with rapamycin was abolished in a mutant lacking the Clr4 methyltransferase, suggesting a possible role for methylation based heterochromatin formation.

Overall this well-written manuscript thoroughly summarizes a great deal of new data that will be of great interest to aging researchers broadly, as well as to *S. pombe* researchers outside of aging.

The methods are clearly explained in detail, and the figures and tables are all very clear and well designed, and all add

greatly to the manuscript overall.

Specific Comments

1. The *S.pombe* strains and cell culture section of the methods, and part of the Isolation of nascent RNA and CLS assay sections, appear to be double-spaced unlike the rest of the manuscript.

Reviewer #2

(Remarks to the Author)

Hirai & Ohta showed that TORC2 factors bind to rDNA and that these binding changes depending on nutritional conditions, and investigated the relation between this phenomenon and the chronological lifespan. Much of the data presented is very clear and compelling. Past reports are also carefully written, and the logical structure is generally easy to understand. However, there are some points that do not meet the above requirements, which are noted below.

Major points

1. (Line 66)

As described in this paper, TORC2 is known to regulate the actin cytoskeleton. Therefore, it will be necessary to consider the possibility that TORC2 also affects actin mRNA, including signal feedback. However, the control for many of the RNA expression studies in this paper uses the expression of actin gene, *act1+*. To demonstrate that this is appropriate, it would be necessary to provide literature or experimental evidence showing that actin mRNA expression is unchanged in TORC2 mutant strains.

2. (Line 123 and 124)

Based on these results, I can agree that TORC2 binds to rDNA and that TORC2 is involved in rRNA transcription regulation. However, is it possible to assert a causal relationship between the two based on the data so far? In other words, if this claim is to be made, it would be better to carry out experiments to deny the possibility that rRNA transcription regulation by TORC2 is unrelated to rDNA binding of TORC2. (Is TORC2 sufficient to control rRNA transcription even without rDNA binding?) Alternatively, a more toned-down claim seems appropriate.

3. (Line 238-240 and 328-337)

Several studies have already shown that TORC2 is involved in chronological lifespan (PMID:23640107, PMID: 23551936 (reference 25), PMID: 33064911). In particular, PMID: 23551936 (reference 25) conducted the exact same experiment in which Δ tor1 was treated with rapamycin and the lifespan was measured, but there is no mention of this in the "Introduction" or "Results". The lifespan of *tor1* Δ in different media has been measured in PMID:23640107 and its discussion has already been done in PMID:33064911. In addition, it appears that lifespan was measured in a culture medium similar to that used in reference 25, but the results are not consistent. If there are any specific changes in measurement, it is better to clearly state what is different from the previous study.

4. Fig. 2g and 2h

Because it seems that no statistical processing has been carried out, there are concerns that the accuracy and reproducibility of this data is low compared to other data. It would be desirable to improve the representation of data to eliminate this concern.

5. Fig. 1b, 3b, etc.

Since Prp3 is known to be involved with Ksg1 (PMID:39476757), which is involved in the TORC2 pathway, is it really suitable as a control? It would be better to use a factor that is not associated with TORC2 as a control or to demonstrate that Prp3 functions as a control in these experiments, including ChIP.

6. Fig. 5 and S5

Why is there no increase in H3K9 methylation in *tor2* mutants or in rapamycin treatment? Are these results consistent with previous findings? I think some explanation is needed for this.

7. Fig. 5f

As a control, lifespan data for a single deletion mutant of *clr4* are needed. If *clr4* Δ has a shorter lifespan than the wild-type strain, in addition to the interpretation described in Line 252-254, it also suggests the possibility that *tor1* and *clr4* independently regulate lifespan. If you cross a long-lived strain with an early-death strain that has different causes, the lifespan of the double mutant strain will be halfway between the early-death and long-lived strains. In addition, in order to discuss heterochromatin and lifespan, it would be better to present not only lifespan data for the *tor1* Δ *clr4* Δ double mutant, but also data on heterochromatin formation in this double mutant.

Minor points

1. (Line 116)

In Pombase, the official name of *pop3+* is written as *wat1+*, so it is better to write it as *wat1+* as well.

2. (Line 196)

At first, it's better not to write Paf1C as an abbreviation, but to write it in a way that makes it easier to understand what it

stands for.

3. (Line 197-199)

"These findings" seems to refer to the cited references in the previous sentence, but please introduce them in the main text as well. There is no mention of glucose starvation in the previous sentence. Provide at least a minimum explanation for this logical development.

4. (Line 349-358)

Lifespan analysis of metformin has also been conducted in fission yeast (PMID: 36941121). It is also introduced in a review (PMID: 38616173). Wouldn't the discussion be more fruitful if these things were taken into account? Meanwhile, although it does not need to be deleted, the discussion of metformin seems to be somewhat less relevant to this paper than other discussions.

5. Fig. 1-4

It seems that the same figure is used in each of Fig. 1-4a. Is there any point in using the same figure multiple times?

Reviewer #3

(Remarks to the Author)

The paper by Hirari and Ohta examines the role of TORC2 in the heterochromatinization of rDNA upon glucose starvation. They show that Tor1, the catalytic subunit of TOR complex 2 (TORC2), and its downstream effector, the Gad8 kinase, accumulate at the rDNA region. The absence of TORC2-Gad8 is associated with the displacement of Paf1C from chromatin upon glucose starvation.

The authors propose that TORC2-Gad8 recruits the Paf1C complex, thereby promoting rRNA transcription at this locus. Upon glucose starvation, when TORC2-Gad8 activity is downregulated (as reported elsewhere), Paf1C is displaced, which in turn promotes rDNA heterochromatin formation. Heterochromatin formation at the rDNA is important for adaptation to poor nutritional conditions (as the authors have shown in previous studies). Heterochromatin formation at the rDNA locus is associated with extended lifespan, similar to the effects of calorie restriction. The authors present data suggesting that loss of TORC2-Gad8 activity leads to an extended lifespan, which is consistent with the idea that heterochromatin formation at the rDNA, but is extremely surprising given previous findings that show rapid loss of viability of TORC2-Gad8 mutant cells in the stationary phase (as detailed below).

Previously, the same authors demonstrated that TORC1 binds chromatin at the rDNA and prevents heterochromatin formation through a mechanism involving Gcn5 and Atf1. Based on their current findings, they propose that TORC1 and TORC2 independently contribute to rDNA heterochromatin formation upon inactivation, ultimately promoting lifespan extension through distinct mechanisms.

The ChIP experiments described in the manuscript are clear and demonstrate the localization of TORC2-Gad8 at a specific chromatin site. The link with the Paf1C complex is intriguing and correlates with previous findings that show physical interactions between Gad8 and Paf1C. However, as detailed below, I have serious concerns regarding their chronological lifespan (CLS) experiments. I also have several comments concerning their interpretations.

Major concerns:

1. Fig. 3c – Since the authors suggest that there is a reduction in the level of Gad8-Flag at the rDNA in Δ tor1 cells, which might reflect a recruitment defect, they should include the overall level of Gad8-FLAG in Δ tor1 cells.
2. Fig. 5 – The results of the CLS experiments are highly surprising and contradict what had been reported for TORC2-Gad8 mutant cells. The authors need to explain why they obtained such drastically different and essentially opposite results compared to findings from two different laboratories. In the current version, they argue that the differences may result from the use of a different medium. However, they are using exact/similar medium that was previously used for such experiments (Rallis et al., 2013; Weisman and Choder, JBC, 2001). As someone who has worked closely with Δ tor1 cells, one of the most notable phenotypes of this strain is its loss of viability in the stationary phase in rich media. If they are using different conditions, they should clarify why their conditions lead to opposite effects, as this explanation is essential for understanding their findings

o I am curious about how precisely the cells are grown—what level of aeration and shaking is used?

o Are they using 96-well plates, which might lead to issues with shaking and homogeneity?

o I am also surprised that wild-type cells in their experiment lose viability only after three days of incubation. This seems like an extremely high rate of death for such cells.

3. Lack of reference to relevant literature – The authors have not cited the paper by Oya et al. (Ekwall, Epigenetics & Chromatin, 2019), which demonstrates the opposite effects of TORC2-Gad8 on Paf1C and heterochromatin formation at the subtelomeric and mating-type regions. Oya et al. suggest that TORC2-Gad8 inhibits Paf1C to promote heterochromatin at subtelomeric regions. It was also shown by the Weisman lab that disrupting Paf1C reverses the lack of heterochromatin at the mating-type or subtelomeric regions, suggesting that TORC2-Gad8 inhibits Paf1C to induce heterochromatin formation at these loci (Cohen et al., 2018). In the current study, the authors suggest that TORC2-Gad8 recruits Paf1C to the rDNA, and loss of TORC2-Gad8 under glucose starvation is required for heterochromatin formation.

o It is possible that Gad8 is present at actively transcribed chromatin, where it interacts with Paf1C. In such regions, its absence is expected to lead to the formation of heterochromatin.

o TORC2 may not localize to subtelomeric (ST) or mating-type (MT) regions, which are normally inactive; thus, its effect on these regions may be indirect, as well as the rescue of de-silencing of TORC2 mutant cells by loss of Paf1C. However, other explanations are also possible.

Further discussion is needed.

4. Discussion – over-interpretation of results – In general, the interpretations are too strong relative to the data presented. For

example:

Page 7 – The authors state:

"Taken together, Paf1C dissociation from rDNA under glucose starvation or in the absence of Gad8 disrupts histone turnover, thereby promoting heterochromatin formation."

The authors do not actually show evidence of histone turnover.

Page 7 – The authors claim:

"These findings suggest that the TORC2-Gad8 pathway suppresses heterochromatin formation by recruiting Paf1C to rDNA loci."

The data do not support the claim that TORC2-Gad8 recruits Paf1C. While previous studies have shown an interaction between Gad8 and Paf1C, further experiments are required to demonstrate a recruitment mechanism.

Minor concerns:

1. Abstract – The abstract would be clearer if it focused on describing the role of TOR2 rather than primarily emphasizing the effects of its inactivation.
2. Page 10, line 311 – The word "despite" does not fit well in the sentence and should be revised for clarity.

Version 1:

Reviewer comments:

Reviewer #2

(Remarks to the Author)

The authors have satisfactorily addressed all of my concerns.

Reviewer #3

(Remarks to the Author)

While the data demonstrating the role of TORC2 in rDNA heterochromatinization are compelling and appear robust, I remain unconvinced by the authors' claim that inactivation of TORC2 leads to a prolonged life span.

In their rebuttal letter, the authors attribute the discrepancies between their findings and those of previous studies to differences in genetic background and medium composition. This is certainly a plausible explanation. However, even within the specific conditions used in their own experiments, the reported effects of tor1 inactivation on life span extension appear minimal.

The small differences observed—particularly at time points when the majority of the culture is already non-viable (for both wild type and mutant strains)—do not, in my view, support strong conclusions about life span extension. The marginal increase in survival under these conditions makes it difficult to distinguish between genuine life span extension and variability associated with late-stage culture decline—possibly reflecting the behavior of a minor subpopulation within the culture.

I would encourage the authors to either provide more convincing evidence—for example, by clearly demonstrating the impact of lysine supplementation on survival—or, preferably, to significantly tone down their conclusions regarding TORC2's role in life span regulation. Doing so would necessitate adjustments to the article's title, abstract, and the interpretation throughout the manuscript. The apparent lack of correlation between TORC2 inactivation promoting rDNA heterochromatinization and its limited or absent effect on life span—perhaps affecting only a small subset of surviving cells—could be explained by considering the multiple roles of TORC2, many of which are likely essential for extended survival.

Version 2:

Reviewer comments:

Reviewer #3

(Remarks to the Author)

I appreciate the considerable effort you have made to revise the manuscript in response to my comments. The changes you describe do reflect a more cautious interpretation, and I am glad to see the title, abstract, and discussion now emphasize association rather than causation.

That said, I remain only partly convinced. In my view, the effect observed is rather minor, and given that it occurs at very low cell numbers, it could reflect variability in late-stage culture decline.

Nonetheless, since the data are clearly presented, I believe it is reasonable to let readers judge for themselves.

Thank you again for your careful revisions.

A point-by-point response to comments by reviewers:

TORC2 inactivation promotes heterochromatin formation in rDNA to extend the chronological lifespan of quiescent fission yeast cells

by Hirai and Ohta

We express our gratitude to the reviewers for their invaluable feedback on our manuscript. In accordance with their recommendations, we have thoroughly addressed all the concerns raised and incorporated substantial additional data, leading to an extensive revision of the manuscript. For further details, kindly refer to our responses (highlighted in black) to the reviewers' remarks and suggestions (presented in *blue italic*). The revised manuscript highlights all changes in **red** for clarity.

Reviewers' comments:

Reviewer #1 (Remarks to the Author):

Summary

*This very interesting article describes the effect of TORC2 inactivation on rDNA heterochromatin formation in the fission yeast *Schizosaccharomyces pombe*. The authors show that TORC2 inactivation leading to rDNA heterochromatin formation also leads to reduced rRNA transcription and increased chronological lifespan. As multiple studies have suggested in several contexts that reducing ribosomal production is linked to increased lifespan in multiple extremely divergent species, this finding could have a broad impact on our understanding of the basic biology of aging.*

The authors first demonstrated that a FLAG tagged version of Tor1, a component of the TORC2 complex, binds rDNA, and that this binding is reduced during glucose starvation, a condition that shows reduced rDNA transcription.

*Next, the authors asked whether lack of TORC2 activity altered rDNA transcription by considering mutants lacking the TORC2 components Tor1, Pop3/Lst8, Sin1, and Ste20/Rictor. In each case, these showed reduced levels of rRNA. This was further confirmed by thiouridine-biotin labeling of newly synthesized rRNA in a *tor1Δ* mutant.*

*Next, the authors considered changes at the rDNA locus. By quantitative PCR they showed a decreased rDNA copy number in the *tor1Δ* mutant, but not one deemed sufficient to fully explain the decreased rRNA. This led the authors to consider heterochromatin formation. They measured H3K9me2 and H3K9me3 levels by ChIP-Seq and found them both increased across the rDNA region in the *tor1Δ* mutant, while the total histone levels appeared unchanged.*

To ask whether Swi6/HP1 bound the increased H3K9me3 seen at the rDNA, the authors

used a GFP-Swi6 along with an mCherry labeled Gar2, which should localize to the nucleolus, the site of the rDNA. They found a slightly increased GFP/RFP colocalization in a *tor1Δ* mutant relative to wild type.

The authors next focused on *Gad8*, a downstream target phosphorylated by TORC2, and showed that *GAD8* deletion led to increased rDNA H3K9me2/me3 and decreased rRNA similar to *tor1Δ*. They also showed that *Gad8-FLAG* accumulated at the rDNA and this was reduced by glucose starvation or *tor1Δ*.

While previous work has suggested that *Atf1* and *Gcn5* dissociation can precede heterochromatin formation at the rDNA, the authors showed that *tor1Δ* and *gad8Δ* do not alter *Atf1* and *Gcn5* localization, suggesting an independent role for TORC2's effects on rDNA heterochromatin.

As *Gad8* has been shown to interact with the *Paf1* complex (*Paf1C*) in telomeric heterochromatin formation, the authors looked at the *Paf1C* component *Leo1* in their context. They showed that *Leo1* localizes to the rDNA and this is reduced in *gad8Δ*, and also that *leo1Δ* leads to increased H3K9 di- and trimethylation and to reduced rRNA.

To consider the relationship between TORC1 and TORC2 the authors combined a TORC1 and TORC2 mutant and showed that the increased H3K9 methylation and reduced rRNA effects were both stronger than in *tor1Δ* alone, and that rapamycin treatment in *tor1Δ* also increased these effects relative to *tor1Δ* alone.

Finally, the authors measured the chronological lifespan of *tor1Δ* mutants and found them greatly longlived relative to wild type, and also found that this effect was further increased by treatment of *tor1Δ* with rapamycin. The effect of *tor1Δ* with rapamycin was abolished in a mutant lacking the *Clr4* methyltransferase, suggesting a possible role for methylation based heterochromatin formation.

Overall this well-written manuscript thoroughly summarizes a great deal of new data that will be of great interest to aging researchers broadly, as well as to *S. pombe* researchers outside of aging.

The methods are clearly explained in detail, and the figures and tables are all very clear and well designed, and all add greatly to the manuscript overall.

(Our reply)

Thank you for reading the manuscript so carefully. We have provided responses to your comments below for your review.

Specific Comments

1. The *S.pombe* strains and cell culture section of the methods, and part of the Isolation of nascent RNA and CLS assay sections, appear to be double-spaced unlike the rest of the

manuscript.

(Our reply)

Thank you for your comment. The double-spacing seems to result from a system issue related to the use of superscript characters. We believe this will be corrected during the production process.

Reviewer #2 (Remarks to the Author):

Hirai & Ohta showed that TORC2 factors bind to rDNA and that these binding changes depending on nutritional conditions, and investigated the relation between this phenomenon and the chronological lifespan. Much of the data presented is very clear and compelling. Past reports are also carefully written, and the logical structure is generally easy to understand. However, there are some points that do not meet the above requirements, which are noted below.

(Our reply)

Thank you very much for reading our manuscript carefully and critically. We greatly appreciate your constructive comments, which were highly insightful. We have carefully considered each of your points and made revisions or clarifications accordingly.

Major points

1. (Line 66)

*As described in this paper, TORC2 is known to regulate the actin cytoskeleton. Therefore, it will be necessary to consider the possibility that TORC2 also affects actin mRNA, including signal feedback. However, the control for many of the RNA expression studies in this paper uses the expression of actin gene, *act1+*. To demonstrate that this is appropriate, it would be necessary to provide literature or experimental evidence showing that actin mRNA expression is unchanged in TORC2 mutant strains.*

(Our reply)

We understand the concern regarding the use of *act1+* as a normalization control, given the potential involvement of TORC2 in regulating actin cytoskeleton dynamics.

To address this, we reviewed previously published transcriptome datasets. In a microarray-based transcriptome study (Schonbrun et al., “TOR complex 2 controls gene silencing, telomere length maintenance, and survival under DNA-damaging conditions”), *act1+* was not among the genes showing significant expression changes (≥ 1.5 -fold) in TORC2 mutant strains. Similarly, in an RNA-seq-based study (Cohen et al., “TOR complex 2 in fission yeast is required for chromatin-mediated gene silencing and assembly of heterochromatic domains

at subtelomeres”), *act1*⁺ was not listed among the differentially expressed genes. Based on this, we believe that *act1*⁺ is a suitable control and have added a brief explanation in the Method section (“Reverse Transcription and Quantitative PCR”).

2. (Line 123 and 124)

Based on these results, I can agree that TORC2 binds to rDNA and that TORC2 is involved in rRNA transcription regulation. However, is it possible to assert a causal relationship between the two based on the data so far? In other words, if this claim is to be made, it would be better to carry out experiments to deny the possibility that rRNA transcription regulation by TORC2 is unrelated to rDNA binding of TORC2. (Is TORC2 sufficient to control rRNA transcription even without rDNA binding?) Alternatively, a more toned-down claim seems appropriate.

(Our reply)

We agree that the current data do not allow us to establish a direct causal relationship between TORC2 binding to rDNA and the regulation of rRNA transcription. In response to your suggestion, we have revised the relevant statement in the Results section (“TORC2 accumulates in the rDNA regions and promotes rRNA transcription”) to tone down the claim and avoid implying direct causality (Paragraph 1 and 2, on page4).

3. (Lien 238-240 and 328-337)

Several studies have already shown that TORC2 is involved in chronological lifespan (PMID:23640107, PMID: 23551936 (reference 25), PMID: 33064911). In particular, PMID: 23551936 (reference 25) conducted the exact same experiment in which $\Delta tor1$ was treated with rapamycin and the lifespan was measured, but there is no mention of this in the "Introduction" or "Results". The lifespan of $tor1\Delta$ in different media has been measured in PMID:23640107 and its discussion has already been done in PMID:33064911. In addition, it appears that lifespan was measured in a culture medium similar to that used in reference 25, but the results are not consistent. If there are any specific changes in measurement, it is better to clearly state what is different from the previous study.

(Our reply)

These are very important and valuable comments. As pointed out, previous studies have reported a shortened chronological lifespan (CLS) in *tor1* Δ cells. We have now cited these studies in the Introduction (Paragraph 2, on page 3) and addressed differences in media conditions in the Discussion, referring to the work by Ohtsuka et al (Paragraph 2 on page 11). To explore the discrepancy between our results and previous reports, we conducted additional CLS assays. In our original study, cells were cultured for two days before

measurement to ensure entry into stationary phase, but we found that the timing of stationary phase entry differs between studies. Therefore, we standardized the initial cell density and measured colony-forming ability from Day -2 (one day after inoculation). Interestingly, *tor1Δ* cells exhibited only about one-third the colony-forming efficiency of wild-type cells at this stage (see below, left), suggesting that the cell density of *tor1Δ* cultures at entry into stationary phase is lower. This observation is consistent with the findings of Weisman and Choder (2001), who described that *tor1Δ* cells have low viability and reduced cell density in stationary phase.

However, when looking at the overall trend in the number of surviving colonies, despite the initially high lethality, a smaller life-shortening effect was observed in *tor1Δ* than in wild type. In contrast, there was a tendency for prolonged survival. In other words, while our data do not contradict the findings of Weisman and colleagues, it suggests that the observed difference lies in the time point at which lifespan extension is evaluated (see below, right).

Another possible explanation is the difference in genetic background: our *tor1Δ* strain carries *leu1⁺*, whereas Weisman et al. used a *leu1-32* strain. Leucine plays an important role in amino acid starvation and the regulation of the TORC1 pathway (in *S. pombe* *tor2*), and a deficiency in this gene may affect the heterochromatinization of rDNA by an alternative redundant pathway. Indeed, based on our unpublished data, *leu1-32* strains exhibited impaired rDNA heterochromatin formation in both nutrient-rich and glucose-starved conditions, which could explain the reduced CLS in their *tor1Δ* background.

Although Rallis et al. used a 972h- background like ours, subtle differences in amino acid concentrations—particularly leucine—might still influence rDNA heterochromatin formation and affect CLS outcomes. We have revised the manuscript to include these points (Paragraph 3, on page 11).

Explanation Figure 1

(Left) Total colony number. Colony counts were measured from Day -2 (defined as one day

after inoculation) through Day 7. Data represent the mean \pm SD (n = 3).

(Right) Cell viability. Colony counts for each day are shown as a percentage relative to the value on Day -2, which was set to 100%. Data represent the mean \pm SD (n = 3).

4. Fig. 2g and 2h

Because it seems that no statistical processing has been carried out, there are concerns that the accuracy and reproducibility of this data is low compared to other data. It would be desirable to improve the representation of data to eliminate this concern.

(Our reply)

In response, we conducted two additional independent experiments to confirm the accuracy and reproducibility of the data. We have updated the figure and the figure legends accordingly in the revised manuscript.

5. Fig. 1b, 3b, etc.

Since Prp3 is known to be involved with Ksg1 (PMID:39476757), which is involved in the TORC2 pathway, is it really suitable as a control? It would be better to use a factor that is not associated with TORC2 as a control or to demonstrate that Prp3 functions as a control in these experiments, including ChIP.

(Our reply)

In response to your suggestion and considering the possibility that *prp3* expression might be affected under glucose starvation or in TORC2-related mutant backgrounds, we instead present the data as %input or normalize it using the *act1* gene as a reference.

6. Fig. 5 and S5

Why is there no increase in H3K9 methylation in tor2 mutants or in rapamycin treatment? Are these results consistent with previous findings? I think some explanation is needed for this.

(Our reply)

Regarding the rapamycin treatment, it has been reported that rapamycin-induced TORC1 inactivation in *S. pombe* is only partial (Hayashi et al., 2007), which may explain why rapamycin treatment alone under glucose-rich conditions influenced neither H3K9 methylation nor rRNA transcription in wild-type cells. However, rapamycin treatment of *tor1* Δ cells induced much denser heterochromatin in the rDNA region and further suppressed rRNA transcription than in rapamycin-untreated *tor1* Δ cells.

As for the *tor2* mutant, while the original temperature-sensitive phenotype is more prominent at 36°C, we performed the experiments at 30°C to avoid confounding effects caused by heat

stress. The partial inactivation of the *tor2-287* single mutant exhibited little increased H3K9 methylation but exhibited reduced rRNA levels in glucose-rich conditions, suggesting that rRNA transcription can also be regulated by heterochromatin-independent mechanisms. Notably, the *tor1Δ tor2-287* double mutant exhibited the greatest increase in H3K9 methylation and the most pronounced reduction in rRNA levels among all strains tested, surpassing the effects observed in either single mutant (Supplementary Fig. 5a, b).

Based on these considerations, we have revised the corresponding description in the manuscript to clarify this point (“Inactivation of both TOR pathways leads to severe heterochromatin formation in rDNA and suppression of rRNA transcription” on page 7 and 8).

7. Fig. 5f

*As a control, lifespan data for a single deletion mutant of *clr4* are needed. If *clr4Δ* has a shorter lifespan than the wild-type strain, in addition to the interpretation described in Line 252-254, it also suggests the possibility that *tor1* and *clr4* independently regulate lifespan. If you cross a long-lived strain with an early-death strain that has different causes, the lifespan of the double mutant strain will be halfway between the early-death and long-lived strains. In addition, in order to discuss heterochromatin and lifespan, it would be better to present not only lifespan data for the *tor1Δclr4Δ* double mutant, but also data on heterochromatin formation in this double mutant.*

(Our reply)

Thank you for your insightful comment. To address your suggestion, we measured the chronological lifespan (CLS) of the *clr4Δ* single mutant. In the *clr4Δ* single mutant, H3K9 methylation in the rDNA region was completely abolished in nutrient-rich conditions, although lifespan was not shortened compared to wild-type. This may be because the heterochromatin levels in the wild-type strain under nutrient-rich conditions fall below the threshold required for lifespan extension, suggesting that its lifespan may already be reduced to a minimal level. More importantly, the lifespan extension observed in the *tor1Δ* mutant treated with rapamycin was abolished in the *tor1Δ clr4Δ* double mutant, where H3K9 methylation was completely abolished, resulting in a shortened lifespan.

Therefore, we concluded that heterochromatin levels seen in quiescent-stage *tor1Δ* cells, with or without rapamycin treatment, are required for the lifespan extension in nutrient-rich conditions. We have incorporated these new data (Supplementary Fig. 5c and d) and interpretations into the revised manuscript to clarify the relationship between heterochromatin and lifespan regulation (paragraph 4 on page 8).

Minor points

1. (Line 116)

In Pombase, the official name of pop3+ is written as wat1+, so it is better to write it as wat1+ as well.

(Our reply)

We have revised the manuscript to refer to the gene as *wat1+*, in accordance with the official nomenclature used in PomBase.

2. (Line 196)

At first, it's better not to write Paf1C as an abbreviation, but to write it in a way that makes it easier to understand what it stands for.

(Our reply)

In the revised manuscript, we have written out the full name of Paf1C (RNA polymerase II-associated factor 1 complex) as its first mention to improve clarity for the reader.

3. (Line 197-199)

"These findings" seems to refer to the cited references in the previous sentence, but please introduce them in the main text as well. There is no mention of glucose starvation in the previous sentence. Provide at least a minimum explanation for this logical development.

(Our reply)

Thank you for your helpful comment. In response, we have revised the relevant paragraph to improve the logical flow and clarity (Paragraph 4, on page 6).

4. (Line 349-358)

Lifespan analysis of metformin has also been conducted in fission yeast (PMID: 36941121). It is also introduced in a review (PMID: 38616173). Wouldn't the discussion be more fruitful if these things were taken into account? Meanwhile, although it does not need to be deleted, the discussion of metformin seems to be somewhat less relevant to this paper than other discussions.

(Our reply)

Thank you for your valuable suggestion. We have now cited the original article reporting the effects of metformin in fission yeast (PMID: 36941121) as well as the recent review summarizing species-dependent variability in metformin's effects on lifespan (PMID: 38616173). Based on these studies we have expanded our discussion to include the relationship between glucose metabolism and lifespan regulation in *S. pombe*, and we have also noted that the effects of metformin are highly dependent on factors such as dosage and timing of administration (paragraph 3 on page 12).

Reviewer #3 (Remarks to the Author):

The paper by Hirari and Ohta examines the role of TORC2 in the heterochromatinization of rDNA upon glucose starvation. They show that Tor1, the catalytic subunit of TOR complex 2 (TORC2), and its downstream effector, the Gad8 kinase, accumulate at the rDNA region. The absence of TORC2-Gad8 is associated with the displacement of Paf1C from chromatin upon glucose starvation.

The authors propose that TORC2-Gad8 recruits the Paf1C complex, thereby promoting rRNA transcription at this locus. Upon glucose starvation, when TORC2-Gad8 activity is downregulated (as reported elsewhere), Paf1C is displaced, which in turn promotes rDNA heterochromatin formation. Heterochromatin formation at the rDNA is important for adaptation to poor nutritional conditions (as the authors have shown in previous studies). Heterochromatin formation at the rDNA locus is associated with extended lifespan, similar to the effects of calorie restriction. The authors present data suggesting that loss of TORC2-Gad8 activity leads to an extended lifespan, which is consistent with the idea that heterochromatin formation at the rDNA, but is extremely surprising given previous findings that show rapid loss of viability of TORC2-Gad8 mutant cells in the stationary phase (as detailed below).

Previously, the same authors demonstrated that TORC1 binds chromatin at the rDNA and prevents heterochromatin formation through a mechanism involving Gcn5 and Atf1. Based on their current findings, they propose that TORC1 and TORC2 independently contribute to rDNA heterochromatin formation upon inactivation, ultimately promoting lifespan extension through distinct mechanisms.

The ChIP experiments described in the manuscript are clear and demonstrate the localization of TORC2-Gad8 at a specific chromatin site. The link with the Paf1C complex is intriguing and correlates with previous findings that show physical interactions between Gad8 and Paf1C. However, as detailed below, I have serious concerns regarding their chronological lifespan (CLS) experiments. I also have several comments concerning their interpretations.

(Our reply)

Thank you very much for your careful and critical reading of our manuscript. We appreciate your insightful summary and particularly value your comments regarding the CLS phenotypes of *tor1* Δ mutant cells. As Reviewer 2 also pointed out, this is indeed a critical issue that warrants careful examination. To address this point, we have conducted additional experiments and revised our discussion to clarify the apparent discrepancies with previous

studies. We hope these revisions adequately address your concerns.

Major concerns:

1. Fig. 3c – Since the authors suggest that there is a reduction in the level of Gad8-Flag at the rDNA in $\Delta tor1$ cells, which might reflect a recruitment defect, they should include the overall level of Gad8-FLAG in $\Delta tor1$ cells.

(Our reply)

We examined the total protein levels of Gad8-FLAG in $tor1\Delta$ cells and found that they were comparable to those in wild-type cells, as determined by western blot analysis. Therefore, the reduced signals at the rDNA observed in the $tor1\Delta$ cells are unlikely to result from decreased protein expression, but rather from impaired chromatin association of Gad8-FLAG. We have added the corresponding data as Fig 3c.

2. Fig. 5 – The results of the CLS experiments are highly surprising and contradict what had been reported for TORC2-Gad8 mutant cells. The authors need to explain why they obtained such drastically different and essentially opposite results compared to findings from two different laboratories. In the current version, they argue that the differences may result from the use of a different medium. However, they are using exact/similar medium that was previously used for such experiments (Rallis et al., 2013; Weisman and Choder, JBC, 2001). As someone who has worked closely with $\Delta tor1$ cells, one of the most notable phenotypes of this strain is its loss of viability in the stationary phase in rich media. If they are using different conditions, they should clarify why their conditions lead to opposite effects, as this explanation is essential for understanding their findings

(Our reply)

These are very important points. As you pointed out, our CLS data differ from previous reports on TORC2-Gad8 mutant strains, and we have carefully considered several possible explanations for this discrepancy.

In our original CLS experiments, we cultured both WT and $tor1\Delta$ cells for two days to allow them to enter stationary phase. However, it is possible that many $tor1\Delta$ cells had already died during this 2-day culture period. To investigate this, we cultured WT and $tor1\Delta$ cells at identical initial cell densities and performed colony formation assays starting the next day (defined as Day -2). We found that, already on Day -2, $tor1\Delta$ cells showed only about one-third of the colony-forming ability compared to WT. This is consistent with previous findings that $tor1\Delta$ cells exhibit a lower cell viability and density in stationary phase (Weisman and Choder, JBC, 2001) (Explanation Fig1, left).

However, when looking at the overall trend in the number of surviving colonies, despite the

initially high lethality, a smaller life-shortening effect was observed in *tor1Δ* than in wild type. In contrast, there was a tendency for prolonged survival. In other words, while our data do not contradict the findings of Weisman and colleagues, it suggests that the observed difference lies in the timespan at which lifespan extension is evaluated (see below, right). Another possible explanation for the discrepancy is differences in genetic background. The *tor1Δ* strains used in Weisman et al. likely have a *leu1-32* background, whereas our strain is *leu1+*. Leucine plays an important role in amino acid starvation and the regulation of the TORC1 pathway (in *S. pombe tor2*), and a deficiency in this gene may affect the heterochromatinization of rDNA by an alternative redundant pathway. Indeed, based on unpublished data, cells with the *leu1-32* background exhibited impaired rDNA heterochromatin formation under both nutrient-rich and glucose-starved conditions. This could result in insufficient rDNA heterochromatinization in stationary phase, leading to reduced viability in the *leu1-32* background.

In contrast, Rallis et al. used *tor1Δ* cells in a 972 h⁻ background, the same as ours. Still, differences in viability were observed, which suggests other factors may be involved. One possibility is variation in the concentration or composition of supplemented amino acids, particularly leucine or lysine. In the methods section of the Rallis et al. paper, they state: "Cells were grown in YES or EMM as described (Roux et al., 2009)." In the Roux et al. study, the YES medium (referred to as YEC in their paper) does not contain lysine, whereas in our experiments, we supplemented the YES medium with lysine. This difference in medium composition may have influenced the observed lifespan phenotypes. Although we have not yet tested whether lysine affects rDNA heterochromatin formation, this is a highly interesting possibility we intend to investigate further.

Importantly, previous studies have also shown that *tor1Δ* cells can exhibit lifespan extension under certain medium conditions. Therefore, we believe our current findings are not contradictory to prior reports but rather reflect differences in medium composition and genetic background. We have revised the manuscript to reflect these points and have added relevant discussion ("Transcriptional repression of rRNA and lifespan extension").

Explanation Figure 1

(Left) Total colony number. Colony counts were measured from Day -2 (defined as one day after inoculation) through Day 7. Data represent the mean \pm SD (n = 3).

(Right) Cell viability. Colony counts for each day are shown as a percentage relative to the value on Day -2, which was set to 100%. Data represent the mean \pm SD (n = 3).

o I am curious about how precisely the cells are grown—what level of aeration and shaking is used?

(Our reply)

Cells were cultured in 20 ml of medium in 100 ml Erlenmeyer flasks, loosely covered with aluminum foil to allow gas exchange. The culture was shaken using a bio shaker with orbital agitation at 30°C. We have added this information to the method section.

o Are they using 96-well plates, which might lead to issues with shaking and homogeneity?

(Our reply)

As mentioned above, each sample was cultured in Erlenmeyer flasks, ensuring sufficient aeration and homogeneity throughout the culture.

o I am also surprised that wild-type cells in their experiment lose viability only after three days of incubation. This seems like an extremely high rate of death for such cells.

(Our reply)

The timing of viability loss can vary depending on how the culture period before reaching stationary phase is defined. As mentioned above, if we define “Day 0” as the day after inoculation when the cells had just entered stationary phase, WT cells began to lose viability around Day 5 to Day 6, which is consistent with previous reports such as Rallis et al.

3. Lack of reference to relevant literature – The authors have not cited the paper by Oya et al. (Ekwall, Epigenetics & Chromatin, 2019), which demonstrates the opposite effects of TORC2-Gad8 on Paf1C and heterochromatin formation at the subtelomeric and mating-type regions. Oya et al. suggest that TORC2-Gad8 inhibits Paf1C to promote heterochromatin at subtelomeric regions. It was also shown by the Weisman lab that disrupting Paf1C reverses the lack of heterochromatin at the mating-type or subtelomeric regions, suggesting that TORC2-Gad8 inhibits Paf1C to induce heterochromatin formation at these loci (Cohen et al., 2018). In the current study, the authors suggest that TORC2-Gad8 recruits Paf1C to the rDNA, and loss of TORC2-Gad8 under glucose starvation is required for heterochromatin formation.

(Our reply)

Thank you for your insightful comment. We have now cited the papers by Oya et al. (Epigenetics & Chromatin, 2019) and Cohen et al. (2018) in the revised Discussion section and addressed the differences between our findings and previous reports (“Heterochromatin formation at rDNA via Paf1C” on pages 9 and 10).

While they suggest that TORC2-Gad8 may inhibit Paf1C to promote heterochromatin formation at sub-telomeric and mating-type regions, our study proposes a distinct mechanism in the rDNA region. Specifically, we found that inactivation of TORC2-Gad8, such as under glucose starvation, leads to the dissociation of Paf1C from the rDNA region, which subsequently induces heterochromatin formation. These differences may reflect region-specific roles of TORC2-Gad8-Paf1C interactions, suggesting that the regulatory mechanisms at rDNA and sub-telomeric loci are distinct.

o It is possible that Gad8 is present at actively transcribed chromatin, where it interacts with Paf1C. In such regions, its absence is expected to lead to the formation of heterochromatin.

(Our reply)

We agree with your interpretation, and we have now added a sentence in the Discussion section to address this possibility (“Heterochromatin formation at rDNA via Paf1C” on pages 9 and 10).

o TORC2 may not localize to subtelomeric (ST) or mating-type (MT) regions, which are normally inactive; thus, its effect on these regions may be indirect, as well as the rescue of de-silencing of TORC2 mutant cells by loss of Paf1C. However, other explanations are also possible. Further discussion is needed.

(Our reply)

In this study, we focused on the rDNA locus; however, as you pointed out, it is certainly

possible that TORC2 indirectly influences the mating-type and subtelomeric regions. For example, TORC2-Gad8, which localizes in the cytoplasm, may phosphorylate Paf1C, thereby preventing its association with these regions. Upon inactivation of TORC2-Gad8, Paf1C could remain in an unphosphorylated state, bind to the mating-type and subtelomeric loci, and potentially interfere with heterochromatin formation.

In contrast, at the rDNA locus, our results raise the possibility of a distinct mechanism in which Gad8 may directly recruit Paf1C to promote transcription. Under conditions of glucose starvation or in Gad8-deficient cells, Paf1C dissociates from the rDNA, resulting in transcriptional repression and heterochromatin formation (“Heterochromatin formation at rDNA via Paf1C” on pages 9 and 10).

4. Discussion – over-interpretation of results – In general, the interpretations are too strong relative to the data presented. For example:

Page 7 – The authors state:

“Taken together, Paf1C dissociation from rDNA under glucose starvation or in the absence of Gad8 disrupts histone turnover, thereby promoting heterochromatin formation.”

The authors do not actually show evidence of histone turnover.

(Our reply)

As you correctly pointed out, we did not directly assess histone turnover in this study. In response, we have revised the sentence to reflect this limitation and avoid making an unsupported conclusion. The revised sentence now reads:

“Taken together, Paf1C dissociation from rDNA under glucose starvation or in the absence of Gad8 may reduce histone turnover, thereby promoting heterochromatin formation.”

Page 7 – The authors claim:

“These findings suggest that the TORC2-Gad8 pathway suppresses heterochromatin formation by recruiting Paf1C to rDNA loci.”

The data do not support the claim that TORC2-Gad8 recruits Paf1C. While previous studies have shown an interaction between Gad8 and Paf1C, further experiments are required to demonstrate a recruitment mechanism.

(Our reply)

Thank you for your comment. As you correctly pointed out, our current data do not provide direct evidence that TORC2-Gad8 recruits Paf1C to rDNA. To address this, we have revised the sentence on page 7 to read:

“These findings raise the possibility that the TORC2-Gad8 pathway suppresses heterochromatin formation, potentially through promoting the association of Paf1C with rDNA

loci."

Minor concerns:

1. Abstract – The abstract would be clearer if it focused on describing the role of TOR2 rather than primarily emphasizing the effects of its inactivation.

(Our reply)

In the revised manuscript, we have modified the abstract to place greater emphasis on the physiological role of TORC2, rather than focusing primarily on the effects of its inactivation.

2. Page 10, line 311 – The word "despite" does not fit well in the sentence and should be revised for clarity.

(Our reply)

Thank you for your comments. We have revised the sentence for clarity by replacing the phrase beginning with "despite" with a clause starting with "although cells are under~" in the revised manuscript.

A point-by-point response to comments by reviewers:

TORC2 inactivation promotes heterochromatin formation in rDNA and prolongs viability of quiescent fission yeast cells

by Hirai and Ohta

We would like to sincerely thank the reviewers for their careful evaluation of our manuscript and their constructive comments.

We have carefully revised the manuscript in response to the reviewers' suggestions, with particular emphasis on tempering the claims regarding TORC2 and lifespan extension as requested.

Below, we provide a point-by-point response.

Reviewer #3

Comment:

Reviewer #3 (Remarks to the Author):

While the data demonstrating the role of TORC2 in rDNA heterochromatinization are compelling and appear robust, I remain unconvinced by the authors' claim that inactivation of TORC2 leads to a prolonged life span.

In their rebuttal letter, the authors attribute the discrepancies between their findings and those of previous studies to differences in genetic background and medium composition. This is certainly a plausible explanation. However, even within the specific conditions used in their own experiments, the reported effects of tor1 inactivation on life span extension appear minimal.

The small differences observed—particularly at time points when the majority of the culture is already non-viable (for both wild type and mutant strains)—do not, in my view, support strong conclusions about life span extension. The marginal increase in survival under these conditions makes it difficult to distinguish between genuine life span extension and variability associated with late-stage culture decline—possibly reflecting the behavior of a minor subpopulation within the culture.

I would encourage the authors to either provide more convincing evidence—for example, by clearly demonstrating the impact of lysine supplementation on survival—or, preferably, to significantly tone down their conclusions regarding TORC2's role in life span regulation. Doing so would necessitate adjustments to the article's title, abstract, and the interpretation throughout the manuscript. The apparent lack of correlation between TORC2 inactivation promoting rDNA heterochromatinization and its limited or absent effect on life span—perhaps affecting only a small subset of surviving cells—could be explained by considering the multiple roles of TORC2, many of which are likely essential for extended survival.

Response:

We thank the reviewer for this important comment. In response to your suggestion, we have carefully revised the manuscript to substantially tone down our statements on chronological lifespan extension. We now describe that TORC2 inactivation is associated with prolonged maintenance of viability in quiescent

cells, as follows:

Title revised:

Original: “TORC2 inactivation promotes heterochromatin formation in rDNA to extend the chronological lifespan of quiescent fission yeast cells”

Revised: “TORC2 inactivation promotes heterochromatin formation in rDNA and prolongs viability of quiescent fission yeast cells ”

We have revised the title to avoid overstating TORC2’s impact on lifespan, ensuring that it more accurately reflects the strength of our evidence.

Abstract revised:

Original statements that implied robust lifespan extension have been replaced with “associated with prolonged viability of quiescent cells”.

This emphasizes association rather than causation.

Results revised:

In response to the reviewer’s comment, we have substantially toned down our interpretation of the TORC2 phenotype in quiescent cells. In the revised Results section, we no longer describe TORC2 inactivation as extending chronological lifespan. Instead, we focus on the observation that TORC2 inactivation is associated with prolonged maintenance of viability in quiescent cells. We have clarified that the effect observed in the *tor1Δ* mutant was moderate, and that a further but still moderate improvement was detected when TORC1 was also inactivated with rapamycin. These changes ensure that our conclusions avoid any overstatement regarding lifespan regulation.

Discussion revised:

In response to the reviewer’s comment, we have revised the Discussion to avoid overstatements regarding lifespan extension. All previous statements implying that TORC2 inactivation “extends lifespan” have been rephrased to indicate that it is “associated with prolonged maintenance of viability in quiescent cells”. Similarly, we now describe the combined TORC1 and TORC2 inactivation as being “associated with prolonged maintenance of viability in quiescent cells” rather than directly causing lifespan extension. These changes ensure that our interpretation reflects an association rather than a causal claim, aligning with the strength of our evidence and the data presented in Figures 5e and 5f.

These revisions ensure that our conclusions are conservative and consistent with the magnitude of the observed effect, fully addressing the reviewer’s concern.

A point-by-point response to comments by reviewers:

TORC2 inactivation promotes heterochromatin formation in rDNA and prolongs viability of quiescent fission yeast cells

by Hirai and Ohta

Reviewer #3

Comment:

Reviewer #3 (Remarks to the Author):

I appreciate the considerable effort you have made to revise the manuscript in response to my comments. The changes you describe do reflect a more cautious interpretation, and I am glad to see the title, abstract, and discussion now emphasize association rather than causation.

That said, I remain only partly convinced. In my view, the effect observed is rather minor, and given that it occurs at very low cell numbers, it could reflect variability in late-stage culture decline.

Nonetheless, since the data are clearly presented, I believe it is reasonable to let readers judge for themselves.

Thank you again for your careful revisions.

Response:

Thank you for your constructive comments and careful reassessment of our revised manuscript.